# An activator of G protein-coupled receptor and MEK1/2-ERK1/2 signaling inhibits HIV-1 replication by altering viral RNA processing

Raymond W. Wong[1,2]*, Ahalya Balachandran[3], Peter K. Cheung[4], Ran Cheng[3], Qun Pan[3,5], Peter Stoilov[6], P. Richard Harrigan[4,7], Benjamin J. Blencowe[3,5], Donald R. Branch[1,2,8,9], Alan Cochrane[3,10]*

1 Department of Medicine, University of Toronto, Toronto, Ontario, Canada, 2 Department of Laboratory Medicine and Pathobiology, University of Toronto, Toronto, Ontario, Canada, 3 Department of Molecular Genetics, University of Toronto, Toronto, Ontario, Canada, 4 British Columbia Centre for Excellence in HIV/AIDS, Vancouver, British Columbia, Canada, 5 Donnelly Centre for Cellular and Biomolecular Research, University of Toronto, Toronto, Ontario, Canada, 6 Department of Biochemistry, West Virginia University, Morgantown, West Virginia, United States of America, 7 Department of Medicine, The University of British Columbia, Vancouver, British Columbia, Canada, 8 Division of Advanced Diagnostics, Infection and Immunity Group, Toronto General Hospital Research Institute, Toronto, Ontario, Canada, 9 Centre for Innovation, Canadian Blood Services, Toronto, Ontario, Canada, 10 Institute of Medical Science, University of Toronto, Toronto, Ontario, Canada

* rw.wong@mail.utoronto.ca (RWW); alan.cochrane@utoronto.ca (AC)

**Data Availability Statement:** Mass spectrometry proteomics data were deposited to the ProteomeXchange Consortium (http://

## Abstract

The ability of HIV-1 to evolve resistance to combined antiretroviral therapies (cARTs) has stimulated research into alternative means of controlling this infection. We assayed >60 modulators of RNA alternative splicing (AS) to identify new inhibitors of HIV-1 RNA processing—a segment of the viral lifecycle not targeted by current drugs—and discovered compound N-[4-chloro-3-(trifluoromethyl)phenyl]-7-nitro-2,1,3-benzoxadiazol-4-amine (5342191) as a potent inhibitor of both wild-type (*Ba-L*, *NL4-3*, *LAI*, *IIIB*, and *N54*) and drug-resistant strains of HIV-1 (IC$_{50}$: ~700 nM) with no significant effect on cell viability at doses tested. 5342191 blocks expression of four essential HIV-1 structural and regulatory proteins (Gag, Env, Tat, and Rev) without affecting total protein synthesis of the cell. This response is associated with altered unspliced (US) and singly-spliced (SS) HIV-1 RNA accumulation (~60% reduction) and transport to the cytoplasm (loss of Rev) whereas parallel analysis of cellular RNAs revealed less than a 0.7% of host alternative splicing (AS) events (0.25–0.67% by $\geq$ 10–20%), gene expression (0.01–0.46% by $\geq$ 2–5 fold), and protein abundance (0.02–0.34% by $\geq$ 1.5–2 fold) being affected. Decreased expression of Tat, but not Gag/Env, upon 5342191 treatment was reversed by a proteasome inhibitor, suggesting that this compound alters the synthesis/degradation of this key viral factor. Consistent with an affect on HIV-1 RNA processing, 5342191 treatment of cells altered the abundance and phosphorylation of serine/arginine-rich splicing factor (SRSF) 1, 3, and 4. Despite the activation of several intracellular signaling pathways by 5342191 (Ras, MEK1/2-ERK1/2, and JNK1/2/3), inhibition of HIV-1 gene expression by this compound could be reversed by pre-treatment with either a G-protein α-subunit inhibitor or two different MEK1/2 inhibitors. These observations demonstrate enhanced sensitivity of HIV-1 gene expression to small changes in host

proteomecentral.proteomexchange.org) via the PRIDE partner repository with dataset identifiers: PXD011079 and DOI 10.6019/PXD011079. All other relevant data are within the manuscript and its Supporting Information files.

**Funding:** This work was supported by a Canadian Institutes of Health Research (CIHR) Fellowship - Priority Announcement (PA): HIV/AIDS Research Initiative (158250) to RWW, http://webapps.cihr-irsc.gc.ca/decisions/p/project_details.html?appId=379093&lang=en. CIHR Doctoral Award - Frederick Banting and Charles Best Canada Graduate Scholarship to RWW, http://webapps.cihr-irsc.gc.ca/decisions/p/project_details.html?appId=283951&lang=en. CIHR Operating Grant - PA: HIV/AIDS Research Initiative (HOP-134065) to AC, http://webapps.cihr-irsc.gc.ca/decisions/p/project_details.html?appId=297384&lang=en. National Institutes of Health (NIH) Grant (R01, EY025536) to PS, https://projectreporter.nih.gov/project_info_history.cfm?aid=9603746&icde=48268061. Ontario Graduate Scholarship to AB, https://www.osap.gov.on.ca/OSAPPortal/en/A-ZListofAid/PRDR019245.html. The funders had no role in study design, data collection and analysis, decision to publish, or preparation of the manuscript.

**Competing interests:** The authors have declared that no competing interests exist.

RNA processing and highlights the potential of modulating host intracellular signaling as an alternative approach for controlling HIV-1 infection.

## Author summary

HIV-1 resistance to current antiretroviral therapies requires new approaches for managing this infection. We assayed over 60 compounds for new inhibitors of HIV-1 RNA processing—an area of the virus lifecycle not targeted by current drugs and primarily under the control of the host cell—and identified compound 5342191 as a potent inhibitor of multiple wild-type and drug-resistant strains of HIV-1. This new inhibitor suppresses production of four essential HIV-1 structural proteins and regulatory factors by reducing expression of most RNAs encoding them and alters synthesis/degradation of at least one of these factors. Inhibition of HIV-1 gene expression by 5342191 resulted from activation of intracellular signaling from G-protein coupled receptors. This study not only exploits a weakness that we previously identified in HIV-1 gene expression but validates and improves the specificity of activating a specific anti-HIV-1 inhibitory signal in the host cell. Omics analyses revealed less than a 0.7% of host processes were changed by 5342191. Future evaluation of this compound or its derivatives in an *in vivo* model of HIV-1 infection will be invaluable for confirming our findings. This study supports the future targeting of host intracellular signaling as a new method for managing HIV-1 infection.

## Introduction

Over 36.9 million people are living with HIV/AIDS [1]. In the absence of a vaccine against HIV-1, cARTs are needed to control the spread and progression of the virus [2,3]. These ARTs target HIV-1 envelope (Env) interactions with its co-receptor (CR) or fusion with the cell membrane, reverse transcriptase (RT), integrase (IN), and protease (PR) [3,4]. However, issues with cARTs, in particular the steady rise (and exponential increase in developing countries) of drug-resistant HIV-1 strains and treatment toxicity, call for further investigation of alternative means of controlling HIV-1 infection [5–9].

Genome-wide siRNA screens, genotyping, and the Human Protein Interaction Database identified 1,254 host genes, including components of several cell signaling pathways, that play important roles in HIV-1 replication [10]. While multiple studies have described diverse kinase signaling pathways modulating HIV-1 entry to its integration [11–22], less is known about which signaling pathways support/impede viral replication at the post-integration stage, especially during HIV-1 RNA processing and protein synthesis [23–25]. Besides promoting HIV-1 infection, phosphatidylinositol-3-kinase (PI3K) activity can regulate HIV-1 gene expression by promoting specific phosphorylation of SR proteins which influence splice site usage of HIV-1 pre-mRNAs [14,26]. Our work determined that cardiotonic steroids (CSs) alter HIV-1 RNA processing to suppress HIV-1 Gag, Env, and Rev expression, an effect correlated with hyperphosphorylation of SRSF3 [27]. Inhibition of HIV-1 gene expression by CSs requires, in part, activation of mitogen-activated protein (MAP) kinase (MAPK)/extracellular signal-regulated kinase (ERK) kinase (MEK) 1/2-ERK1/2 by a mechanism that is independent of the toxic/arrhythmogenic properties of this family of drugs [28]. However, the narrow therapeutic index of CSs makes this family of drugs unattractive candidates to repurpose as ARTs [27–29]. Thus, while MEK1/2-ERK1/2 signaling may play a supportive role during early stages

of viral infection (pre-integration) [14,19,21,22,30,31], we revealed that signaling through this pathway can negatively impact the expression of integrated HIV-1 provirus [28]. An approach to exploit this apparent weakness could lead to alternative means of controlling HIV-1 infection and, perhaps, evolve into a new class of ARTs.

In this study, we assayed over 60 modulators of RNA splicing for inhibitors of HIV-1 expression that act at the post-integration level. We identified compound 5342191 as a potent inhibitor of HIV-1 gene expression. Distinct from other inhibitors of HIV-1 replication, this benzoxadiazole blocks the expression of several essential HIV-1 (Gag and Env) and regulatory proteins (Tat and Rev) with no significant impact on total protein synthesis of the cell. 5342191 also potently inhibits the replication of several HIV-1 strains, including drug-resistant strains, in CD4+ T cells with no reduction in cell viability at concentrations tested. Loss in expression of four HIV-1 structural/regulatory proteins upon 5342191 treatment is correlated with a substantial decrease in accumulation and cytoplasmic transport of viral US/SS RNAs encoding them (described in S1 Fig). In contrast, 5342191 changes less than 0.7% of host AS events and abundance of RNAs at doses required to suppress HIV-1 gene expression. Subsequent assays determined that, while 5342191 induced the activation of several MAPK signaling pathways [MEK1/2 and c-Jun N-terminal kinase (JNK) 1/2/3], only a subset of these were required for its antiviral effect. While inhibitors of MAPKs (JNK1/2/3 and p38), Src, $Ca^{2+}$ flux, or epidermal growth factor receptor (EGFR) signaling had no effect, inhibitors of MEK1/2 or G-protein coupled receptors (GPCRs) were able to restore the expression of both Gag and viral RNAs in the presence of 5342191. In contrast to CSs that act via inhibiting the $Na^+/K^+$-ATPase on the cell surface, our data supports a model in which 5342191 inhibits HIV-1 gene expression by promoting MEK1/2-ERK1/2 signaling through activation of GPCRs at the cell membrane. Given that inhibition of HIV-1 gene expression by both 5342191 and CSs requires activation of MEK1/2-ERK1/2 signaling, our observations suggest that this signaling pathway plays an important role in regulating HIV-1 replication [28]. The findings of this study support the targeting of a host intracellular signaling pathway as an alternative method for controlling HIV-1 infection.

## Results

### Compound 5342191 suppresses the expression of essential HIV-1 structural and regulatory proteins

To identify small molecule inhibitors of HIV-1 replication, compounds from the Chembridge library able to alter splicing of SMN2 RNA (from P. Stoilov) were evaluated for their effect on HIV-1 gene expression using HeLa cells transduced with a Tet-ON HIV-1 *LAI* provirus (rtTA-HIV-Δ*Mls*). Modifications made to the integrated HIV-1 *LAI* provirus rendered its gene expression dependent upon addition of doxycycline (Dox) [32,33]. Treatment of these cells with 5342191 for 24 h (Fig 1A) strongly inhibited HIV-1 gene expression in a dose-dependent manner (inhibitory concentration of 50%, IC$_{50}$: 750 nM, Fig 1B), with no discernible effects on cell viability. This compound had a similar effect over 1 day on HIV-1 gene expression in a CD4+ T-cell line transduced with a *NL4-3*-derived provirus (24ST1NLESG, IC$_{50}$: 750 nM, Fig 1C) [34]. The capacity of 5342191 to inhibit HIV-1 gene expression in both cell lines suggests a conserved inhibitory mechanism.

To define the basis for this response, expression of HIV-1 structural and regulatory proteins were evaluated in the context of rtTA-HIV-Δ*Mls* HeLa cell line [33]. Western blot analysis demonstrated that 5342191 blocked expression of key HIV-1 structural proteins (Fig 1D): Gag polyprotein (p55), matrix-capsid (MA-CA, p41), and CA (p24) and Env polyprotein (gp160) and processed product (gp120). In addition, this compound caused a loss in vital HIV-1

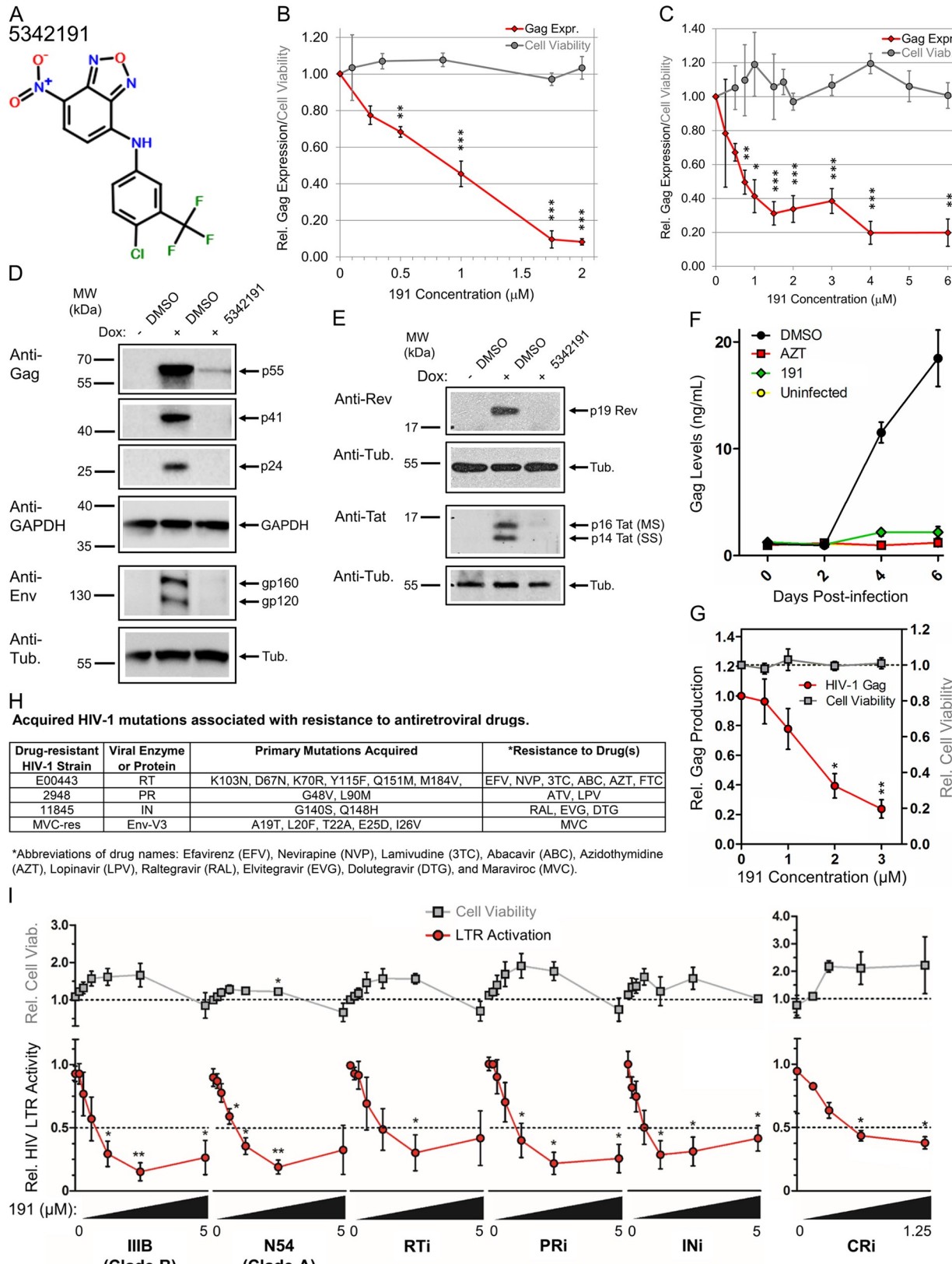

H

**Acquired HIV-1 mutations associated with resistance to antiretroviral drugs.**

| Drug-resistant HIV-1 Strain | Viral Enzyme or Protein | Primary Mutations Acquired | *Resistance to Drug(s) |
|---|---|---|---|
| E00443 | RT | K103N, D67N, K70R, Y115F, Q151M, M184V, | EFV, NVP, 3TC, ABC, AZT, FTC |
| 2948 | PR | G48V, L90M | ATV, LPV |
| 11845 | IN | G140S, Q148H | RAL, EVG, DTG |
| MVC-res | Env-V3 | A19T, L20F, T22A, E25D, I26V | MVC |

*Abbreviations of drug names: Efavirenz (EFV), Nevirapine (NVP), Lamivudine (3TC), Abacavir (ABC), Azidothymidine (AZT), Lopinavir (LPV), Raltegravir (RAL), Elvitegravir (EVG), Dolutegravir (DTG), and Maraviroc (MVC).

**Fig 1. 5342191 inhibits HIV-1 gene expression and replication.** (**A**) 5342191 chemical structure. HeLa rtTA-HIV-Δ*Mls* (**B** and **D-E**) or CD4$^+$ 24ST1NLESG T cells (**C**) were treated with indicated concentrations or IC$_{90}$ (2 μM) of 5342191 (191), or DMSO (control) only for 4 h prior to Dox or PMA (+) induction (resp.) of HIV-1 expression for 20 h. Each treatment contained equal concentrations of DMSO solvent. After ~24 h, cell supernatants were harvested for (**B-C**) p24$^{CA}$ ELISA of HIV-1 Gag expression (black diamonds) and XTT assay of cell viability (gray circles; n ≥ 4–5, mean, s.e.m.) while (**D-E**) cell lysates (~30 μg) were analyzed by immunoblot for expression of (**D**) HIV-1 structural proteins: Gag (p55, p41, and p24) and Env (gp160 and gp120), (**E**) viral regulatory factors: Rev (p19) and Tat (p16 and p14), and internal loading controls: GAPDH or α-tubulin (n ≥ 4–6, mean, s.e.m.). Position of molecular weight (MW) standards were marked adjacent to each gel/blot. Lanes in (**D-E**, resp.) were cropped/assembled from the same blots (S2A and S2B Fig). (**F-G**) Primary CD4$^+$ T cells (PBMCs) were infected or left uninfected (yellow circles) with HIV-1 *Ba-L*, treated with 3 μM 5342191 (191, green diamonds), 3.7 μM AZT (red boxes), or DMSO only (black circles), which was partially replenished with fresh drug & media after 4 days, and cell supernatant harvested every 2 days for p24$^{CA}$ ELISA to monitor effects on (**F**) HIV-1 growth over 0–6 days and (**G**) dose-response of 0–3 μM of 5342191 on HIV-1 replication (red circles) and cell viability (gray boxes, by trypan blue exclusion) on day 6. Data from (**F-G**) are n = 4 from 1 donor, representative of 4 different donors, mean, s.e.m. (**H-I**) CEM-GXR cells were infected with WT (*IIIB* or *N54*) or RT inhibitor (i), PRi, INi, or CRi-resistant strains of HIV-1 described in (**H**), treated with 0, 0.15, 0.3, 0.6, 1.25, 2.5, or 5 μM of 5342191 (gradient bar, except CRi: 0–1.25 μM), and their effects on (**I**) HIV-1 LTR activation (red circles) and cell viability (gray boxes) quantified from GFP fluorescence and live-cell counts, respectively, by flow cytometry after 3 days of culture (n ≥ 3, except CRi: n ≥ 2–3, mean, s.e.m.). Dotted-black lines in (**G** and **I**) mark 100% cell viability or IC$_{50}$. All results are relative and statistically compared to treatment with 0 μM of compound.

regulatory factors (Fig 1E): Rev and both isoforms of Tat (p16 and p14, described in S1 Fig). Rev is critical in mediating nuclear export of incompletely-spliced (US/SS) HIV-1 RNAs to the cytoplasm while Tat is essential as a transactivator of viral transcription [35–38]. These results (and data on HIV-1 RNAs and SR proteins described below) are in mark contrast to previous HIV-1 RNA-processing inhibitors reported, suggesting a novel mechanism of action [27,28,33,39–43].

## 5342191 inhibits replication of wild-type (WT) and drug-resistant HIV-1 strains

To assess whether 5342191 is an effective inhibitor of replication-competent HIV-1 in the context of primary CD4+ T cells, peripheral blood mononuclear cells (PBMCs) were activated, infected with a R5 lab strain of HIV-1 (*Ba-L*), and viral replication monitored in the presence/absence of this compound (Fig 1F and 1G). Treatment of HIV-1-infected PBMCs with 5342191 (partially replenished once on day 4) impeded viral growth over 6 days of culture (Fig 1F) in a dose-dependent manner (Fig 1G). 5342191 also inhibits the replication (measured by Tat-dependent LTR activation) of other WT HIV-1 strains, i.e. clade B (*IIIB* and *NL4-3* from Fig 1C) and, clade A (*N54*), and strains resistant to one of the four classes of ARTs in CD4$^+$ CEM-GXR T cells cultured for 3 days (IC$_{50}$: ~750 nM, Fig 1H and 1I) [44], with no significant changes in cell viability at concentrations tested.

## Inhibitor 5342191 alters the accumulation of HIV-1 RNAs

To understand the basis for the reduction in HIV-1 protein expression upon 5342191 addition, we evaluated whether the compound induced changes in viral RNA accumulation. qRT-PCR of RNAs from 5342191-treated HeLa rtTA-HIV-Δ*Mls* cells (Fig 2A and 2B) revealed that 5342191 shifts the accumulation of HIV-1 RNAs: reducing both US and SS mRNA levels by ~60% but increasing MS mRNA abundance by ~140% relative to DMSO (+) controls. A similar effect was observed for 5342191 on HIV-1 RNAs in CD4+ 24ST1NLESG T cells (Fig 2C). However, the changes in viral RNA levels only partially account for the effect of 5342191 on the expression of HIV-1 proteins (Fig 1D and 1E). Fluorescent *in situ* hybridization (FISH) analysis of HIV-1 genomic (US) RNA localization in HeLa rtTA-HIV(Gag-GFP) cells determined that 5342191 treatment for 24 h drastically reduced cytoplasmic accumulation of US RNAs (Fig 2D), with residual signal detected only in the nucleus compared to control. These

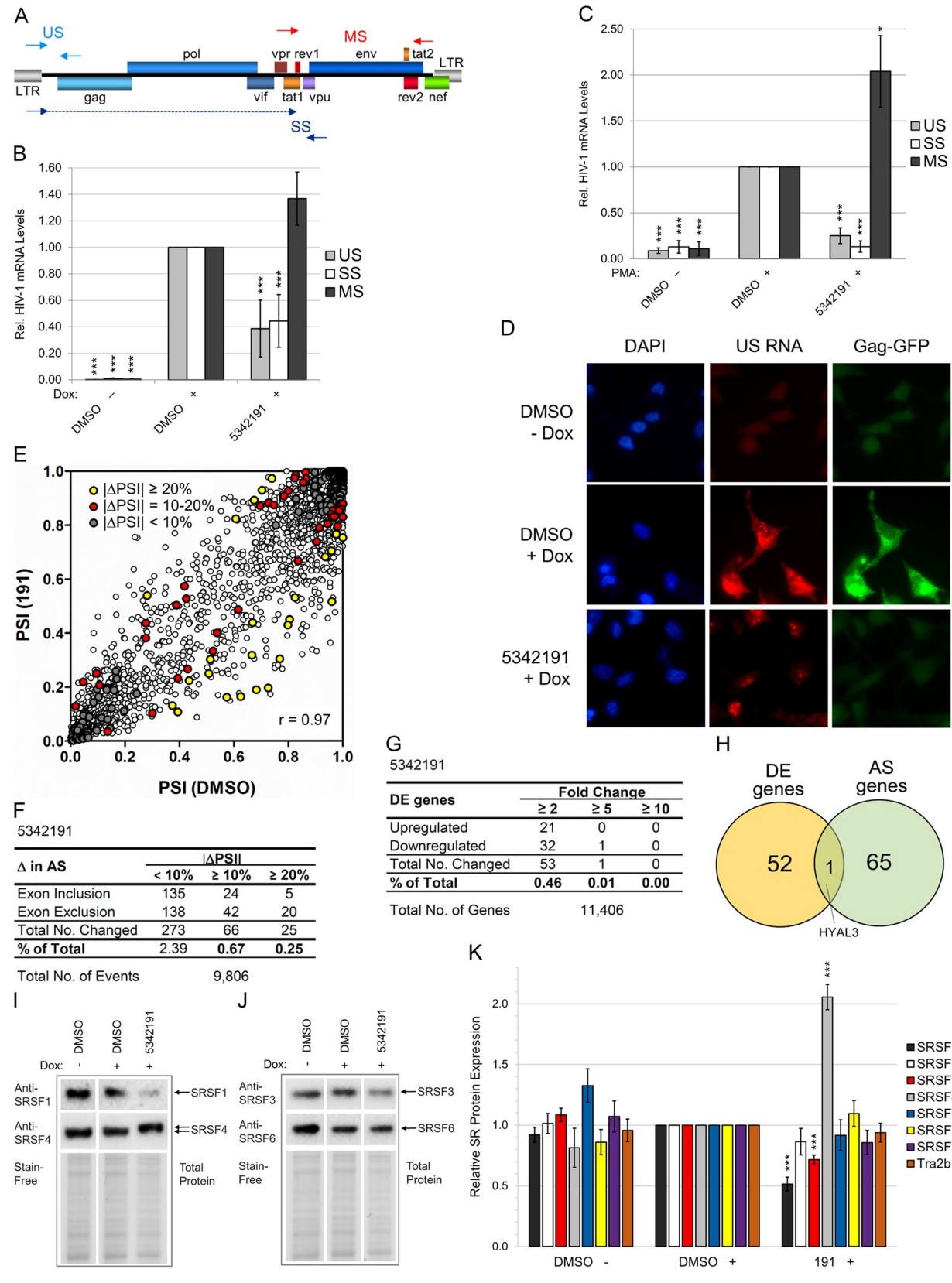

**Fig 2. 5342191 affects HIV-1 RNA processing and expression/modification of SR splicing factors with limited perturbation of AS and expression of host RNAs.** HeLa rtTA-HIV-Δ*Mls* (**A-B** and **E-H**), 24ST1NLESG T (**C**), and HeLa rtTA-HIV(Gag-GFP) cells (**D**) and (**I-K**) were treated with/without IC$_{80}$-IC$_{90}$ of 5342191 (2, 4, 2, and 2.5 μM, resp.) and Dox per Fig 1 and assayed as follows. (**A-C**) Quantitation of the relative expression of HIV-1 US (gray), SS (white), and MS RNAs (black) in cells by qRT-PCR. (**A**) Diagram of the HIV-1 genome indicating position of primers used in amplification. Solid arrow heads denote start while dashed-lined arrows represent exon coverage. (**B-C**) Graph of RNA levels quantified from HeLa rtTA-HIV-Δ*Mls* and 24ST1NLESG T cells (resp.) treated with/without 5342191 (n ≥ 3, mean, s.e.m.). Results are relative and statistically compared to DMSO (+) for each RNA class. (**D**) Trafficking of US RNAs (labeled with Texas Red) detected by FISH (representative of n ≥ 3). Nuclei were detected by DAPI stain, Gag-GFP expression by GFP, and images captured at 630x magnification. (**E-H**) RNA-Seq quantifying the AS of host RNAs (mean PSI) from 9,806 exon inclusion/exclusion events examined and DE genes [mean fold change (Δ)] from 11,406 host RNAs detected from 5342191 or DMSO-treated cells (S1 and S3 Tables; n = 2, mean). (**E**) Scatterplot of PSIs displaying differences in AS between 5342191 (y-axis) and DMSO (x-axis) with significant ΔPSIs (p <0.05) indicated by colored circles as follows: <10% (gray), 10–20% (yellow), and ≥ 20% (red). (**F**) Total number and percentage (%) of AS events altered (ΔPSI ≥ 10% and 20%), (**G**) total number and % of DE genes changed (≥ 2, 5, and 10 fold), and (**H**) Venn diagram of the AS (≥ 10%) and DE genes (≥ 2 fold) affected in common. (**I, J,** and **S5 Fig**) Representative immunoblots and (**K**) graph quantifying the accumulation (and modification) of endogenous SR proteins from lysates of treated cells (~30 μg, n ≥ 3, mean, s.e.m.). Results are relative and statistically compared to DMSO (+). β-actin (**B-C**) and Stain-Free-labeled total proteins (**I-K**) served as internal loading controls for normalization of RNA and protein data, respectively.

results indicate that 5342191 not only alters HIV-1 US RNA accumulation but also movement of these RNAs to the cytoplasm, consistent with the observed loss in Rev expression (Fig 1E).

## 5342191 induces limited alterations to the host transcriptome

Given the 5342191-induced changes in HIV-1 RNA accumulation (Fig 2A–2D), we explored the effect of this compound on host RNA processing and abundance in HeLa rtTA-HIV-Δ*Mls* cells by both RNA-Seq and RT-PCR. A total of 9,806 exon inclusion/exclusion events were analyzed with confidence from 18,611 AS events detected by RNA-Seq (S1 Table), representing ~10% of 100,000 intermediate to high-abundance AS events reported for multi-exon human genes [45,46]. Out of the 9,806 AS events analyzed (S1 Table), 5342191 treatment of cells for 24 h changed the extent of exon inclusion (percent spliced in, PSI) of only 66 events (0.67%) by ≥ 10% compared to DMSO (+) while only 25 events (0.25%) had ≥ 20% perturbations (r = 0.97, Fig 2E and 2F). In parallel, a RT-PCR assay of 70 AS events (S3A Fig and S2 Table) revealed a high degree of correlation with data obtained from RNA-Seq (r = 0.83, S3B Fig and S1 Table). In addition, the expression of 11,406 genes were analyzed with confidence from 19,847 transcripts detected by RNA-Seq (S3 Table) which represents 95–114% of the 10,000–12,000 proteins expressed in all cells (or 67–71% of 16,000–17,000 transcripts/proteins reported for the human transcriptome/proteome)[47]. 5342191 altered the abundance of 53 mRNAs (0.46%) by ≥ 2 fold (Fig 2G) from 11,406 genes detected and analyzed by RNA-Seq (S3 Table) but applying a higher cut-off of ≥ 5 fold reduced the total number of these alterations to only 1 mRNA (0.01%). Comparisons between 5342191-induced changes in AS and these differentially expressed (DE) genes at a ≥ 10% and 2-fold cut-off, respectively (Fig 2F and 2G), found only one host gene affected at both levels (HYAL3, Fig 2H). Collectively, these results indicate that 5342191, at doses which potently suppress HIV-1 gene expression, does not act through general perturbation of AS or gene expression of the host cell but rather through selective effects on a subset of cellular RNAs (Figs 1 and 2). This subset of alterations induced by 5342191 are well tolerated by the cell as indicated by a lack of cytotoxicity at concentrations tested in over 5 different cell lines/types (Fig 1B and 1C, S4 Fig, and signaling experiments below) and, especially, in prolonged cultures of transformed and primary T cells (Fig 1I and 1G, resp.).

## 5342191 modulates the expression and modification of a subset of SR splicing factors

To determine whether alterations in HIV-1 RNA accumulation by 5342191 treatment (Fig 2B) could be the result of changes in the abundance or activity of splicing factors, we evaluated the

effect of this compound on SR protein levels and phosphorylation in HeLa rtTA-HIV(Gag-GFP) cells after 24 h. Comparison of 5342191 versus DMSO-treated cells (Fig 2I–2K and S5 Fig) revealed no significant change in the abundance of most SR proteins examined with the exception of SRSF1 (~2-fold decrease to 50%), SRSF4 (2.1-fold increase to 210%), and SRSF3 (~30% decrease to 70%), all of which have been shown to affect HIV-1 RNA processing and expression [27,48–55]. Furthermore, reduced migration of SRSF4 (Fig 2I) is consistent with changes in its phosphorylation.

## 5342191 reduction of Tat protein levels occurs without affecting total protein synthesis of the cell

While reduced Gag expression (Fig 1) reflects changes in accumulation and subcellular distribution of US RNAs, reduced Tat and Rev expression upon 5342191 addition (Fig 1E) contrasts with the level of HIV-1 MS RNAs (Fig 2B). RT-PCR analysis of splice site usage within the MS RNAs from HeLa rtTA-HIV-Δ*Mls* cells (S6A–S6C Fig) did not reveal specific changes in the level of any mRNAs that could account for reductions in Tat and Rev levels in 5342191-treated cells. Using the surface sensing of translation (SUnSET) technique [56], immunoblot analysis of nascent peptides determined that 5342191 treatment of HeLa rtTA-HIV-Δ*Mls* cells for 24 h resulted in no significant change in the rate of new protein synthesis compared to DMSO (+, or –Dox, S7A–S7C Fig). However, addition of a proteasome inhibitor (MG132) for 8 h prior to harvest resulted in partial rescue of Tat in 5342191-treated cells (Fig 3A and 3B). In contrast, MG132 addition did not restore the expression of other HIV-1 proteins (Gag or Env) in the presence of 5342191. These results indicate that Tat synthesis occurs in the presence of 5342191 and its degradation is mediated by the host proteasome.

## 5342191 causes only 0.02–0.34% change to the host proteome

The possibility that 5342191 may be affecting Tat expression by altering its synthesis/degradation (Fig 3A and 3B) raised the question of whether host proteins were similarly affected. To test this possibility, HeLa rtTA-HIV-Δ*Mls* cells were treated with 5342191, 9147791 (a previously identified HIV-1 RNA processing inhibitor) [40], or DMSO (+/- Dox) and abundance of proteins analyzed by liquid chromatography-tandem mass spectrometry (LC-MS/MS) using tandem mass tags (TMT) [57]. From volcano plots (S8A–S8C Fig) of the abundance and statistical significances of 5,326 proteins identified by TMT LC-MS/MS (S4 Table), a cut-off of $\geq$ 1.5-fold change was selected to identify the proteins which were substantially affected above the normal ranges of each treatment (Fig 3C–3E and S9A–S9E Fig). Although no proteomics analysis covers all proteins, the total number of proteins we detected (S4 Table) represents 44–53% of the 10,000–12,000 proteins present in all cells (or 31–33% of 16,000–17,000 proteins reported for the human proteome) [47]. Compared to DMSO (+) reference, 5342191 induced little change in the abundance of 5,326 proteins identified (r = 0.990, Fig 3C). 5342191 altered the abundance of only 18 proteins (0.34%) in cells by $\geq$ 1.5-fold and only 1 protein (0.02%) by $\geq$ 2.0 fold (Fig 3D and listed in S5 Table). In contrast, 9147791 and DMSO (-) treated cells had more than or near double the number and amplitude of perturbations compared to reference (S9A and S9B Fig, r = 0.980 and 0.981, resp.): 40 (0.75%) and 31 proteins (0.58%) showed a $\geq$ 1.5-fold change (S9C and S9D Fig, resp., and listed in S5 Table). 5342191 altered levels of only 7 and 6 proteins in common with 9147791 and DMSO (-), respectively (Fig 3E and S5 Table). Of the proteins affected, 2 are HIV-1 proteins, Gag and Tat (S5 Table). From the total number of cellular proteins affected by >1.5 fold in these experiments, only 4, 8, and 5 proteins showed significantly altered expression (p <0.05) from reference in cells treated with 5342191, 9147791, and DMSO (-), respectively (top left and top right quadrants of

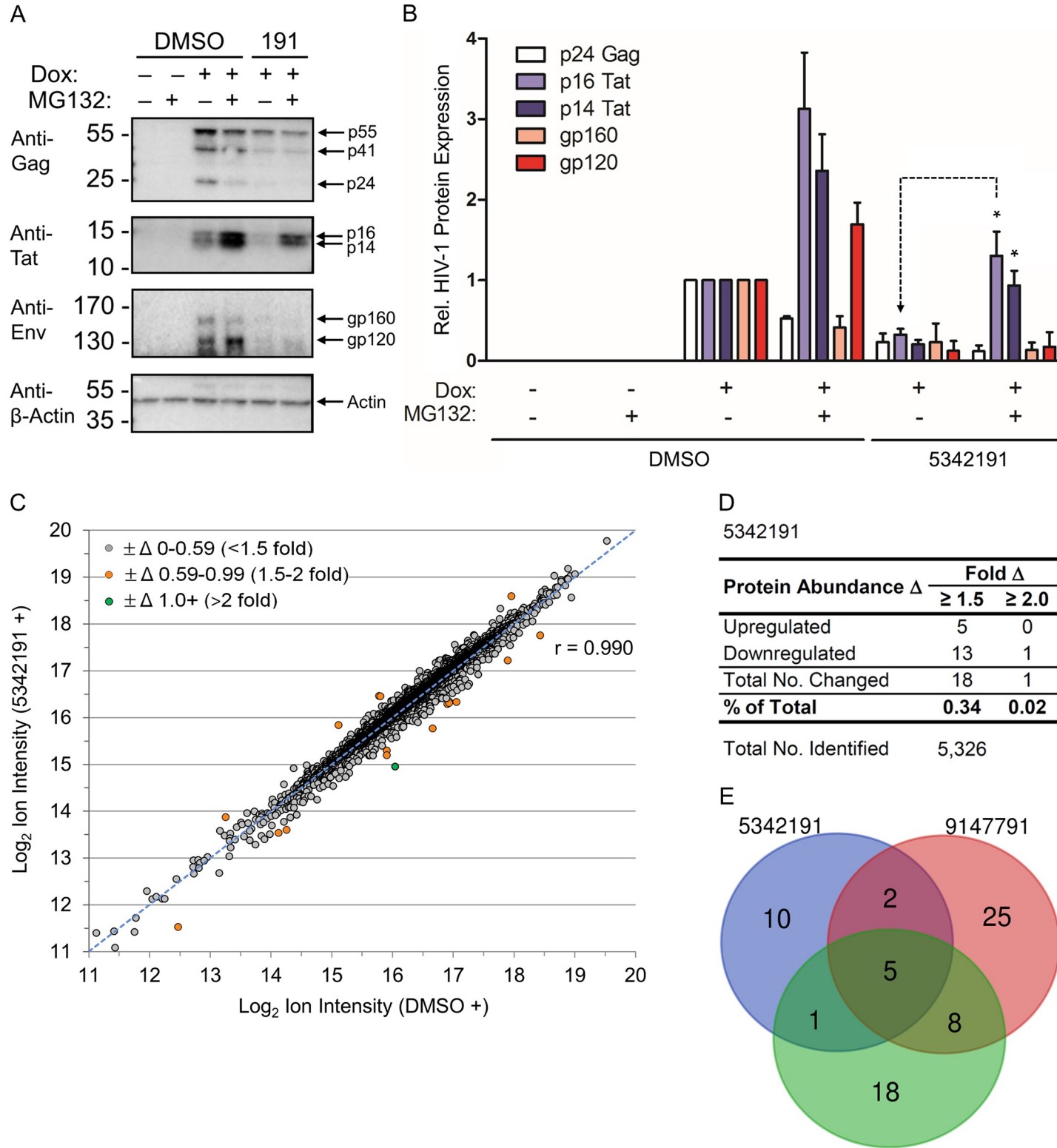

**Fig 3. 5342191 reduction of Tat expression is reversed by inhibition of the proteasome and has limited effects on the abundance of host proteins.** HeLa rtTA-HIV-ΔMls cells were treated with/without 2 μM 5342191 and Dox per Fig 1 and viral/host protein levels quantified as follows. (**A-B**) Cells treated with/without 5342191 and Dox for 24 h and with/without addition of proteasome-inhibitor MG132 8 h prior to harvest were immunoblotted for HIV-1 CA (p24 Gag, white), p16 (purple) and p14 Tat (blue), and gp160 (pink) and gp120 Env (red) from lysates (~30 μg). (**A**) Representative immunoblots and (**B**) multi-bar graph of these results (n ≥ 3, mean, s.e.m.).

Results in (**A-B**) are relative to DMSO (+) and cells treated with 5342191 and MG132 were statistically compared to treatments with this compound and no MG132 as illustrated by a black-dashed line for one of the targets. (**C-E**) Averaged normalized ion intensities ($\log_2$) from 5,326 proteins identified by TMT LC-MS/MS (S4 Table) were analyzed as follows. (**C**) Proteins from 5342191 (y-axis, n = 3) and DMSO (+)-treated cells (x-axis, n = 2) were plotted to illustrate differences in abundance between treatments with the levels of each change depicted by colored circles as follows: <1.5-fold Δ (+/- 0–0.59, gray), 1.5-2-fold Δ (+/- 0.59–0.99, orange), and >2-fold Δ (+/- $1.0^+$, green). Degree of correlation (r, Pearson) and a blue-dashed line representing a drug with no impact on protein abundance are shown. (**D**) Total number and percentage (%) of proteins affected by 5342191 that were ≥ 1.5 and 2-fold changed from control. (**E**) Venn diagram comparing proteins with ≥ 1.5-fold Δ between 5342191, 9147791, and DMSO (-) from data in (**D**) and S9C and S9D Fig, respectively.

S8A–S8C Fig and S5 Table). With the exception of 9147791 (which enriched for proteins involved in biosynthesis of amino acids and carbon metabolism), none of the treatments demonstrated an enrichment for any pathways (S6 Table).

## 5342191 inhibits HIV-1 gene expression through intracellular signaling

CSs suppress HIV-1 gene expression through inhibition of the $Na^+/K^+$-ATPase on the cell surface, resulting in activation of ERK1/2 and other MAPKs (JNK and p38), PI3K-AKT, and PLC and increase intracellular $Ca^{2+}$ concentration ($[Ca^{2+}]_i$) [28]. Given that the 5342191-induced alterations in viral RNA processing are similar to those observed following CS treatments (Figs 1–3), we explored whether 5342191 acts by modulating similar signaling pathways [27,28]. Consistent with the hypothesis that 5342191 acts by inducing signaling within the cell, exposure of HeLa rtTA-HIV(Gag-GFP) cells to 5342191 for only ~4 h (with removal from cell media) was enough to suppress HIV-1 gene expression (S10 Fig). Furthermore, we observed that 5342191 activates ERK1/2 (target of MEK1/2), JNK1/2/3, and MAPK-activated protein kinase-2 (MAP-KAPK-2 or MK-2, a target of p38 MAPK; Figs 4A and S11A). In contrast to CSs (e.g. ouabain) [28], 5342191 did not increase $[Ca^{2+}]_i$ (Fig 4B) or p38 MAPK activation, despite activating MK-2, relative to DMSO (+) control (Fig 4A). Furthermore, 5342191 has no significant effect on the levels of reactive oxygen species (ROS, a secondary messenger of Ras and PI3K-AKT and potential activator of Ras and MAPKs) or $Na^+/K^+$-ATPase in the cell (S13A–S13C Fig), where inhibition or depletion of the latter could result in a similar antiviral response as CSs [28]. These results suggest that 5342191 induces MAP/MAPK signaling by a mechanism distinct from that of CSs.

To assess which, if any, of the 5342191-induced changes in signaling are required for its effect on HIV-1 expression, HeLa rtTA-HIV(Gag-GFP) cells were pretreated with kinase inhibitors for 3 or 15 h prior to addition of 5342191 and Dox and then monitored for Gag-GFP expression. Pre-treatment of cells with inhibitors of MEK1/2 (MEKi #1: U0126 or MEKi #2: Selumetinib/AZD6244), p38α/β/β2 (p38i: SB203580), JNK1/2/3 (JNKi: SP600125), a chelator of intracellular $Ca^{2+}$ ($[Ca2+]i$: BAPTA-AM), or inhibitor of $Ca^{2+}$ influx via the $Na^+/Ca^+$-exchanger (NCXi: KB-R7943) were confirmed to block 5342191 or ouabain-induced activation of MAP/MAPK kinases or $[Ca^{2+}]_i$ influx (Fig 4A and 4B, S11B and S11C Fig, and/or alongside our previous study [28]) with little effect on cell density at concentrations applied (S12B and S12D Fig). Pre-treatment with MEKi #1, but not other MAPK or $Ca^{2+}$ influx inhibitors, partially rescued HIV-1 gene expression (47%) in 5342191-treated cells compared to control (no kinase inhibitor and 5342191, Fig 4C, S11D and S12A and S12C Figs). Consistent with the effects of MEKi #1 (with one off-target kinase reported) [58], pre-treatment of cells with another inhibitor of MEK1/2 activity (MEKi #2: Selumetinib, which blocks ERK1/2 activation, Fig 4A and S11C Fig) resulted in almost complete rescue (77%) of HIV-1 gene expression in 5342191-treated cells compared to DMSO (+, Fig 4C and S11C Fig) [59].

## 5342191 activates GPCR signals to suppress HIV-1 gene expression

To understand how 5342191 activates the MEK1/2-ERK1/2 pathway (Fig 4A and 4C, S11A–S11D, S12A and S12C Figs), we examined what other signaling pathway(s) were required for

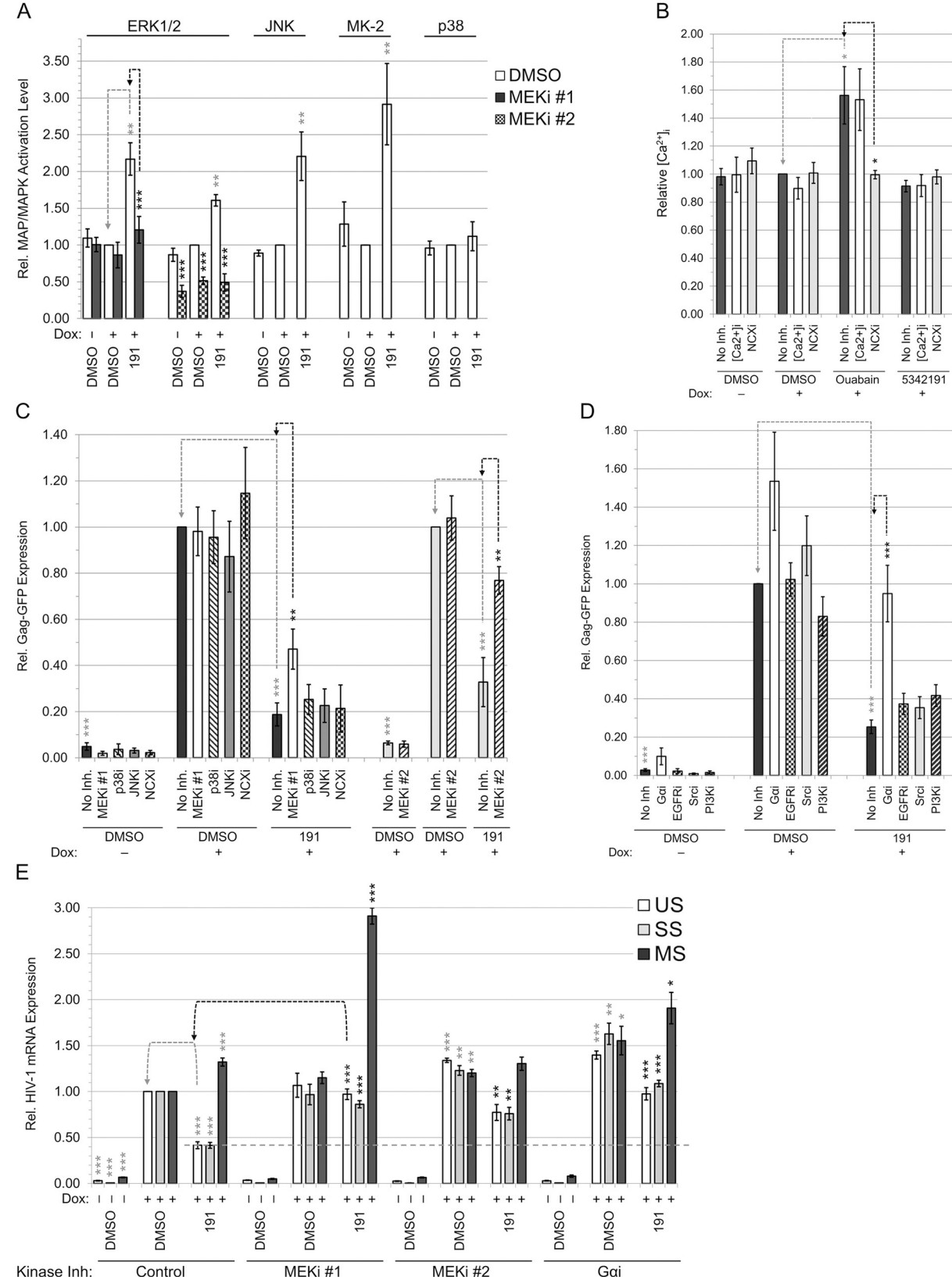

**Fig 4. Inhibition of HIV-1 gene expression by 5342191 is reversed by blocking MEK1/2-ERK1/2 or GPCR signaling.** HeLa rtTA-HIV (Gag-GFP) cells were pretreated with/without a kinase inhibitor prior to treatment with 2 μM 5342191 or DMSO and Dox per Fig 1 to determine the signaling pathway(s) involved by monitoring kinase activation and/or recovery of Gag-GFP expression in cell lysates. Cells were pretreated with/without a kinase inhibitor (and confirmed to block each kinase) as follows: MEKi #1 (12 μM U0126), MEKi #2 (5 μM Selumetinib), JNKi (1.25 μM SP600125), or p38i (15 μM SB203580) overnight for ~15 h (**A** and/or alongside previous study [28]); [Ca2+]i (5 μM BAPTA-AM) or NCXi (5 μM KB-R7943) for 3 h (**B**); and Gαi (~10 μM BIM-46187,), EGFRi (120 nM PD158780,), Srci (350 nM Herbimycin A), or PI3Ki (10 μM LY294002) for 3 h (Fig 5A on Ras activity and/or S12E Fig via EGF effects on HIV-1 expression). Impact of each of these inhibitor combinations on cell density were monitored in S12B, S12D, and S12F Fig. Tubulin or Stain-Free-labeled total proteins served as internal loading controls and for normalization of these data. (**A**) Graph quantitating the activation levels of ERK1/2 (with/without MEKi #1 or #2), JNK1/2/3, MK-2, and p38 in treated cells (n ≥ 3–5, mean, s.e.m.) as observed and described in representative blots provided in S11A–S11C Fig. (**B**) Graph quantifying relative $[Ca^{2+}]_i$ from fluorescence of Fura Red AM loaded into cells treated with/without 5342191 (n ≥ 4, mean, s.e.m.). Ouabain treatment served as a positive control in this assay [28]. (**C-D**) Graphs quantitating rescue of Gag-GFP expression in treated cells by detecting fluorescence from reducing SDS-PAGE (n ≥ 3–4, mean, s.e.m., and initially from live/fixed-plated cells of S12A and S12C Fig) as described in representative gels of S11C–S11F Fig. Note that data on the right side of (**C**) contains a second experiment set for MEKi #2. (**E**) Graph quantifying the level of HIV-1 RNAs in treated cells by qRT-PCR (n ≥ 3–4, mean, s.e.m.). Assay was performed as described in Fig 2A–2C. All results from (**A-E**) are relative to DMSO (+) and no kinase inhibitor (as well as for each RNA class for qRT-PCRs). Statistical comparisons were performed as illustrated by gray/black-dashed lines (and asterisks) for one of the targets. A horizontal gray-dashed line marks the level of US and SS RNAs in 5342191-treated cells with no kinase inhibitor.

its inhibition of HIV-1 gene expression. Analysis of the lysates of 5342191-treated HeLa rtTA-HIV(Gag-GFP) cells revealed an increase in Ras GTPase activity (Ras binding to Raf) relative to DMSO (+, Fig 5A), suggesting that MEK1/2-ERK1/2 activation may result from altered activity of pathways regulating this factor. In support of a role for Ras/Raf/MEK1/2/ERK1/2 signaling in the regulation of HIV-1 gene expression, exogenous expression of N-Ras [WT or oncogenic (12D)] inhibited expression of HIV-1 Gag (Fig 5B and 5C) [60]. In contrast, dominant-negative (17N) N-Ras, with reduced guanine nucleoside exchange factor (GEF) activation and Raf binding, displayed reduced inhibition of HIV-1 expression compared to WT N-Ras (Fig 5B and 5C). Furthermore, the extent of inhibition of HIV-1 gene expression by each type of N-Ras correlated with their degree of ERK1/2 activation (Fig 5B). However, WT Ras overexpression did not induce the same extent of alterations in HIV-1 RNA accumulation as those observed upon 5342191 addition (Fig 5D), indicating that increased Ras activity alone may not account for the complete response to this compound. Furthermore, treatment of cells with several other kinase inhibitors (i.e. EGFRi and Srci) blocked the increase in Ras activity upon 5342191 addition (Fig 5A) without restoring HIV-1 Gag expression in the presence of this compound (Fig 4D, S11E and S11F Fig and S12E and S12F Fig). Of the pathway inhibitors tested, only a pan-Gα protein inhibitor (Gαi), which blocks GPCR signaling (as well as increased ERK1/2 activation, S11G Fig), restored Gag-GFP expression (95%) in cells treated with 5342191 to levels comparable to control-treated cells (Fig 4D, S12E and S12F Fig, and S11E and S11F Fig).

To validate that inhibitors of GPCR or MEK1/2 signaling could block the effects of 5342191 treatment on viral RNA processing (Fig 5E), we examined whether they reversed the effects of this compound on accumulation HIV-1 RNAs by qRT-PCR. In all instances, kinase inhibitors that rescued Gag expression (MEKi #1, MEKi #2, and Gαi, Fig 4C and 4D) also increased US and SS HIV-1 RNAs to levels at par with those of controls (+ DMSO, Figs 4E and 5E). MEKi #1 and Gαi addition also increased the level of MS HIV-1 RNAs (2.9 and 1.9 fold, resp.).

## Discussion

Development of many antiviral agents have focused on inhibitors that selectively target HIV-1 encoded functions (RT, IN, PR, and Env) [3,4]. While this approach has met with success, the rapid evolution of this virus has resulted in the selection of variants resistant to most of these agents [5,6,9,61]. As an alternative approach, we focused on targeting host cellular processes essential for HIV-1 replication. A benefit of this approach includes a potentially greater barrier

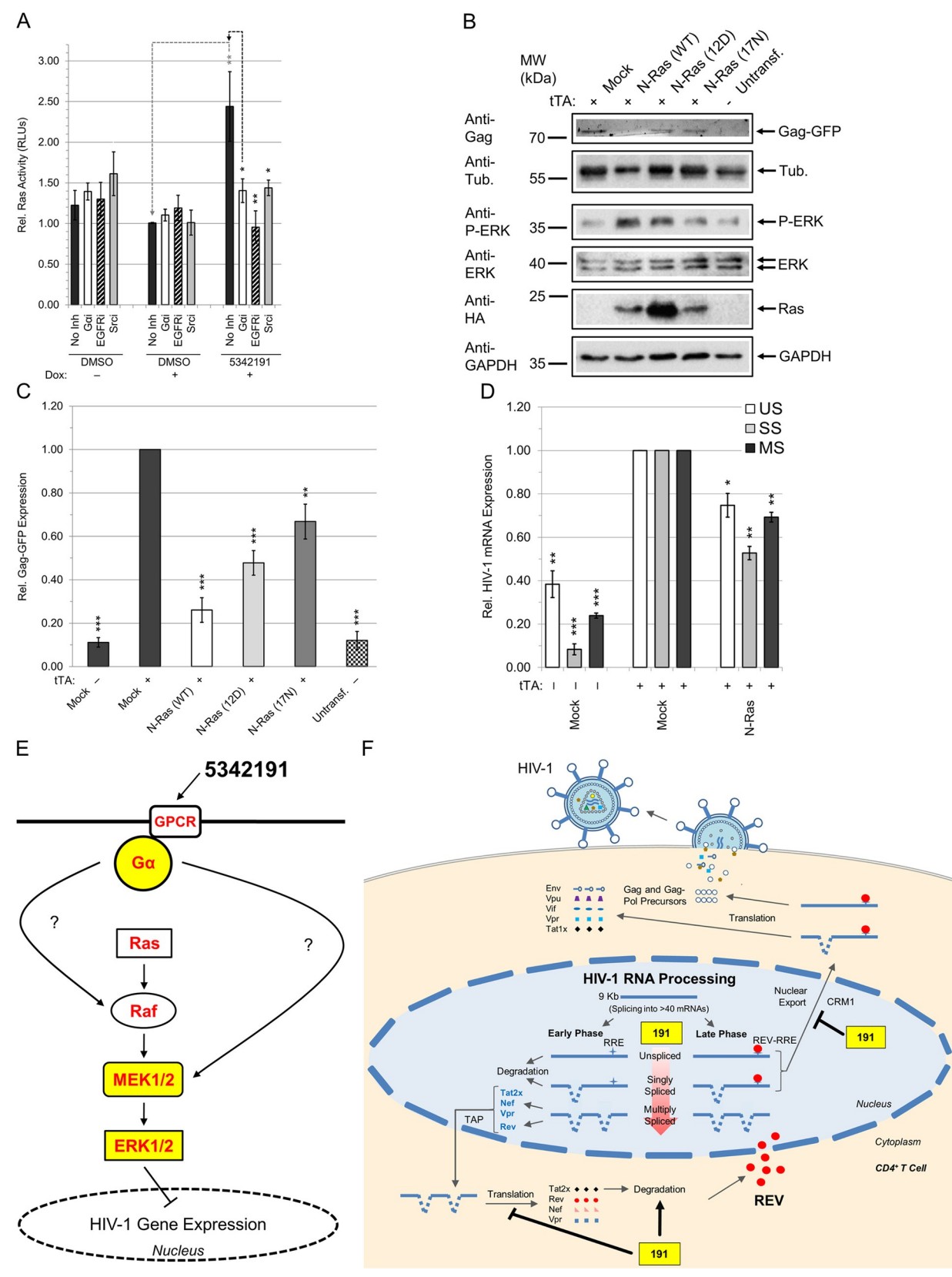

**Fig 5. Exogenous expression of variants of the small G protein, N-Ras, inhibits HIV-1 gene expression.** (**A-D**) HeLa rtTA-HIV(Gag-GFP) cells were (**A**) treated with/without 5342191 per Fig 1 or (**B-D**) transfected with WT, oncogenic (12D), or dominant-negative (17N) N-Ras, or mock plasmid and co-transfected with (+) tTA (or without, -) to activate HIV-1 expression. After 48 h, cell lysates were analyzed as follows. (**A**) Graph of the relative Ras activity (relative RLUs) in lysates (45 μg) of cells treated with/without 5342191 quantified by Ras-GTP ELISA (n ≥ 3, mean, s.e.m.). (**B**) Representative gel of Gag-GFP expression detected by GFP fluorescence from reducing SDS-PAGE and representative immunoblots of ERK1/2 (phospho and total), HA-tagged N-Ras, and α-tubulin/GAPDH assayed from 30 μg of cell lysates (representative of n ≥ 3–5). (**C-D**) Graphs quantifying the level of expression of (**C**) HIV-1 Gag described in (**B**, n ≥ 3–5, mean, s.e.m.) and (**D**) HIV-1 RNAs detected by qRT-PCR as described in Fig 2A and 2B (n ≥ 3, mean, s.e.m.). (**E-F**) Models of the antiviral signal (**E**) and impact (**F**) of 5342191 (191) on HIV-1 gene expression in cells. This study describes a new small molecule inhibitor (5342191) which, in a similar manner as CSs, activates (arrows) MEK1/2-ERK1/2 signaling to inhibit (block arrows) HIV-1 replication (**E-F**). Unlike CSs which function via the $Na^+/K^+$-ATPase, 5342191 activates Gα proteins of GPCRs at the cell membrane (**E**). Although 5342191 also stimulates JNK and MK-2 (a target of p38 MAPK), it achieves suppression of HIV-1 gene expression without initiating p38 MAPK or NCX-mediated $[Ca^{2+}]_i$ flux observed with CSs (**E**, detailed in S15 Fig). 5342191 treatment of cells decreases accumulation of HIV-1 incompletely-spliced (US/SS) RNAs (and increases MS RNAs, progressively-red arrow; **F**) through activation of GPCR and MEK1/2-ERK1/2 signals described in (**E**) and impedes buildup of essential viral regulatory proteins (Tat and Rev, **F**). This results in blocked expression and cytoplasmic transport (loss of Rev) of US and SS HIV-1 RNAs which encode key HIV-1 structural (Gag and Env) and regulatory proteins (p14 Tat) that are essential for HIV-1 replication (**F**). Addition of this compound also reduces accumulation of HIV-1 regulatory factors in cells by altering translation of MS RNAs and/or degradation of viral regulatory factors such as Tat (**F**). This new inhibitor (5342191) achieves potent suppression of HIV-1 replication with little/no changes in cell viability, alternative splicing and expression of host RNAs, or nascent synthesis and abundance of host proteins.

to development of drug resistance [41] while its feasibility is supported by the existence of a number of well-tolerated therapies that target host processes to treat other diseases (i.e. dementia, congestive heart failure, hypertension, cardiac arrhythmias, diabetes, etc.) [62–64]. As a test of this approach, we examined small molecule modulators of SMN2 RNA AS and identified 5342191 as an inhibitor of HIV-1 replication. While targeting components of the core splicing apparatus (requiring coordinated functioning of many host proteins for removal of introns from RNAs) may be toxic [65], modulation of factors regulating splice site use, such as SR proteins or heterogeneous nuclear ribonucleoproteins, may not have as severe of an effect on the host [27,39,41,42,66]. Our characterization of 5342191 as a potent inhibitor of HIV-1 replication (Fig 1) validates this hypothesis. Although this compound has been described to weakly inhibit microsomal prostaglandin E synthase-1, the concentrations required to suppress HIV-1 gene expression ($IC_{50}$: 700 nM-1.8 μM, Fig 1) is well below those required to inhibit this enzyme (50 μM for 71% reduction), supporting the hypothesis that 5342191 modulation of HIV-1 RNA accumulation (Fig 2B–2D) is a novel activity unrelated to its effect on this enzyme [67]. Consistent with this conclusion, treatment of cells with 5342191 altered the abundance/modification of a subset of SR proteins known to regulate HIV-1 RNA processing (Fig 2K) [27,42,43,48,50,54,55,68]: increasing SRSF4 levels (abundance and modification) by 2.1 fold (to 210%) and decreasing SRSF1 and SRSF3 abundance by 2 fold and 30% (to 50% and 70%, resp.) relative to controls, without altering SRSF 6, 7, 9, and Tra2α levels. 5342191 treatment results in reduced accumulation of viral structural (Gag/Env) and regulatory (Tat/Rev) proteins essential for HIV-1 replication (Fig 1, modeled in Fig 5F). Consistent with the critical roles played by both Tat and Rev in HIV-1 replication [35–37], 5342191 inhibited the replication of multiple HIV-1 strains (*Ba-L*, *LAI*, *NL4-3*, *IIIB*, and *N54*), including ones resistant to existing ARTs (Fig 1). These dose-response curves have an average $IC_{50}$ of ~700 nM except those for *Ba-L* infected PBMCs ($IC_{50}$: 1.8 μM, Fig 1G), a likely result of donor to donor variation, lower compound replenishment frequency to cell media (every 4 days compared to 3–4 days), and length of cell culture (6 days compared to ≥ 8–10 days) [27,28,39]. Supporting this presumption, treatment of chronic HIV-infected PBMCs from a clinical patient over 20 days with 5342191 demonstrated a similar dose response to HIV-1 infected T-cell lines ($IC_{50}$: 750 nM) [69]. These results demonstrate that this compound is not only useful as a probe for studying the regulation of HIV-1 RNA processing but, with further development into a more targeted approach, 5342191 and/or its derivatives could prove useful in salvage therapies and/or in combination with existing anti-HIV-1 drugs [70].

While reduced Gag, Env, and p14-Tat expression (Fig 1) can be attributed to reduced abundance and nuclear sequestration of HIV-1 US/SS RNAs (Figs 5F and 2B–2D), the basis for the loss of p16 Tat and Rev (Fig 1E) was unclear given that 5342191 had no significant effect on the accumulation and splice site usage of MS viral RNAs (Fig 2B and 2C and S6 Fig, resp.). The partial rescue of Tat protein upon MG132 addition to cells (Fig 3B) confirmed that HIV-1 MS mRNAs are being translated in the presence of the compound, indicating that 5342191 either selectively reduces translation of Tat RNAs, promotes degradation of this protein, or both. The failure of MG132 to elicit similar restoration of Gag or Env expression (Fig 3B) is possibly due to the compound acting to enhance viral RNA processing from unspliced to spliced RNAs (Fig 2B and 2C) or is an indirect effect on US/SS RNA transport due to a loss of Rev (Figs 2D and 1E). Consistent with the latter interpretation, the effect of 5342191 on HIV-1 RNAs is comparable to an inhibitor of Rev-mediated RNA export, leptomycin B (LB, S14 Fig), which significantly reduces US and SS RNA accumulation with little effect on MS RNA abundance [71].

Inhibition of HIV-1 gene expression by 5342191 in cells (24 h, Figs 1 and 2) occurred with alterations of less than 0.7% of the AS and expression of host RNAs (Fig 2F and 2G), nascent protein synthesis of cells (S7 Fig), and abundance of host proteins (Fig 3D). In comparison with other HIV-1 RNA processing inhibitors, 5342191 induces fewer alterations to host AS events (0.25–0.67% change and r = 0.97 compared to DMSO, resp., in Fig 2E and 2F) than those detected for the CS digitoxin (20.6%) and ABX464 (r = 0.89) by RNA-Seq or chlorhexidine (8.1%) by exon microarray [41,72,73]. 5342191-induced changes in AS convert to only 0.46% of host genes becoming DE by $\geq$ 2 fold in cells (Fig 2G) which is lower than those reported for other HIV-1 RNA processing inhibitors at this threshold (and same cell type): 1C8 (0.95%) and 9147791 (0.75%) [40,42]. Although not previously reported for any other HIV-1 RNA processing inhibitors, 5342191 perturbs the abundance of only 0.02–0.34% of host proteins by $\geq$ 1.5–2.0 fold (Fig 3D) which is over 2.2–6.5-fold lower than the changes observed in 9147791-treated cells at this cut-off (0.13–0.75%, S9C Fig). Upon comparison of the transcriptomics and proteomics data (S9E and S9F Fig), we found 0 hits in common between the 48 RNAs altered $\geq$ 2.0 fold and 18 proteins changed $\geq$ 1.5 or 1.3 fold upon addition of 5342191. Although integration of transcriptomics and proteomics data are generally known to have low correlation [74], our results are also confounded by a low number of total changes detected in the cells by each of these omics studies (Figs 2G and 3D). Consistent with these results, there were no substantial effects of 5342191 at doses required to suppress HIV-1 gene expression on the metabolism/proliferation/viability of multiple cell lines and primary cells used in this study (Fig 1, S12 Fig and S4 Fig). These findings suggest that HIV-1 RNA processing is highly sensitive to selective changes in splicing factor activity (Fig 2K), consistent with recent work by our group and others using other compounds that modulate HIV-1 RNA abundance (8-azaguanine, 5350150, 9147791, ABX464, and 1C8) [39–42,44].

In exploring the mechanism by which 5342191 inhibits HIV-1 replication or gene expression (Fig 5E and detailed in S15 Fig), we determined that 5342191 induced two of the same signaling pathways activated by CSs (ERK1/2 and JNK1/2/3, Fig 5E and Fig 4A and 4B). 5342191 did not induce alterations in $[Ca^{2+}]_i$ or activate p38 MAPKs (Fig 4A and 4B), suggesting that this compound does not affect HIV-1 by inhibiting (or reducing expression of, S13B and S13C Fig) the $Na^+/K^+$-ATPase—primary target of CSs—but rather activates the MEK1/2-ERK1/2 pathway by a distinct mechanism (Figs 5E, 4A and 4C and S11G) [75]. Inhibition of either PI3K or AKT signaling failed to reverse suppression of HIV-1 gene expression by 5342191 (Fig 4D and S15 Fig) [28]. Of the kinase pathways activated by 5342191 (Fig 4A and 4B), only inhibition of the MEK1/2-ERK1/2 pathway, using two highly-specific MEK1/2 inhibitors, restored HIV-1 Gag and US/SS RNA expression (Fig 4C and 4E), suggesting that one or more of the ~200 substrates of this pathway could contribute to the observed response (Fig 5E and 5F) [76]. Supporting this conclusion, at least two genes downstream of ERK1/2, cAMP response element-binding protein (*CREB-2/ATF4*)

and CCAAT/enhancer-binding protein beta (*C/EBPβ*), were detected by RNA-Seq to increase in expression by 1.5- and 1.4-fold (resp.) in cells treated with 5342191 (S15 Fig and S3 Table). CEM-GXR T cells treated for 3 days with this compound also induced a trending increase in viable cell counts (Fig 1I), which likely resulted from ERK1/2 induction of genes promoting cell proliferation and survival [76,77]. Our observation that other activators of the MEK1/2-ERK1/2 pathway (anisomycin and CSs) also inhibit HIV-1 gene expression supports our hypothesis that the effect of 5342191 on HIV-1 replication could be mediated through this signaling cascade [28].

The 5342191-induced increase in Ras activity (Fig 5A) raised the possibility that Ras activation could be responsible for the reduction in HIV-1 gene expression (Fig 5E and detailed in S15 Fig). While pre-treatment with pan-Gα, EGFR, or Src inhibitors blocked the activation of Ras by 5342191 (Fig 5A), only the pan-Gα subunit inhibitor rescued HIV-1 Gag and US/SS RNA expression in the presence of this compound (Fig 4D, 4E and 5E). This observation indicates that Ras activation alone cannot account for the response to 5342191. In agreement with this conclusion, we observed that, while overexpression of WT Ras suppressed HIV-1 Gag expression (Fig 5C), it was not accompanied by changes in viral RNA accumulation similar to that of 5342191-treated cells (Figs 5D and 2B). However, the recovery of both Gag and US/SS viral RNA expression in the presence of the pan-Gα inhibitor (Fig 4D and 4E) raises the possibility that 5342191 is an agonist of a GPCR receptor (Fig 5E). Supporting this assumption, activation of ERK1/2 by this compound was also blocked by the pan-Gα subunit inhibitor (S11G Fig) in a similar manner as MEK1/2 inhibitors. Further delineation of this response will require identification of which of ~1000 GPCRs on the cell surface and their corresponding G α-subunit intracellular signaling mediators ($G\alpha_{i/o}$, $G\alpha_{q/11}$, $G\alpha_{12/13}$, or $G\alpha_s$) are necessary for the effect of 5342191 on HIV-1 expression (S15 Fig) [78]. Although $G\alpha_q$ signals have been reported to be critical for fusion of R5 viruses (pre-integration) in U87 glioblastoma cells, there are no reports of this G protein subclass or its GPCR in regulating HIV-1 gene expression during the post-integration stage of the virus lifecycle [17]. Thus, inhibition of HIV-1 replication by 5342191 in a GPCR-dependent manner is a novel finding, suggesting that targeting of this receptor class could potentially complement current cARTs inhibiting virus-encoded functions [3]. In addition, this report complements recent studies demonstrating that non-receptor tyrosine kinases can be targeted by small molecules to block HIV-1 entry and integration [79], further highlighting the importance of intracellular signaling before and after integration of the provirus and new means of controlling its replication.

This study identifies the benzoxadiazole, 5342191, as a novel inhibitor that suppresses HIV-1 replication by reducing the abundance of US and SS viral RNAs and blocking the accumulation of four essential viral structural and regulatory proteins (Fig 5F). This response is achieved with minimal effects on cell viability, AS and expression of host RNAs, or synthesis and abundance of host proteins. Examination of 5342191's antiviral mechanism identified that, unlike CSs which inhibit $Na^+/K^+$-ATPase function, this compound requires components of GPCR signaling and the MEK1/2-ERK1/2 pathway (Fig 5E) to inhibit HIV-1 gene expression but does not activate p38 MAPKs or $[Ca^{2+}]_i$ flux associated with the toxicity of CSs. This study highlights the potential of targeting a host intracellular signaling pathway, with minimal side effects to the host, as a new alternative method for controlling HIV-1 gene expression (Fig 5E and 5F). The fact that 34% of current FDA-approved drugs target various GPCRs [80] indicate that targeting this class of receptors and the signaling pathways that they regulate is a feasible approach for managing HIV-1 infection.

## Materials and methods

### Ethics statement

Written informed consent was obtained from volunteer blood donors in accordance with guidelines for conducting biomedical research and experimental protocols approved by the

University of Toronto HIV Research Ethics Board or St. Michael's Hospital (Toronto, Canada) research ethics committee (REB #12–378).

## Cells used in identifying new inhibitors of HIV-1 replication

**Inducible HIV-1 cell lines**. Two Tet-ON HIV-1 cell lines containing a HIV-1 (*LAI*) provirus [rtTA-HIV-Δ*Mls* or rtTA-HIV(Gag-GFP)] in HeLa cells, activatable by Dox or tTA, were modified and generated (resp.) to assay compounds for effects on HIV-1 gene expression as previously described [28,33]. After treating cells for 4 h with pre-diluted compound/drugs, HIV-1 expression was induced with 2 μg/mL of Dox. Equal concentrations of DMSO solvent were present in each experiment. After ~20 h, cells and media were harvested to monitor effects of treatments as described below. CD4+ T-cell line, 24ST1NLESG (from J. Dougherty), was treated with compounds as described above but induced by phorbol 12-myristate 13-acetate (PMA) for 24 h as previously performed [27,34,39]. CD4+ T-cell line, CEM-GXR, which expresses GFP upon activation of a Tat-dependent LTR promoter after HIV-1 infection, were treated with compounds and assayed after 3 days as previously described [40,44]. Compound 5342191 and 9147791 were purchased from ChemBridge Online Chemical Store (**www.hit2lead.com**) while 3'-azido-3'-deoxythymidine (AZT) was from Sigma-Aldrich (#A2169). These compound/drugs were solubilized in DMSO. **HIV-1-infected T cells**. PBMCs were isolated from healthy volunteer blood donors (uninfected with HIV), leukophoresed, and stored at -80˚C until use. PBMCs were activated with 2 μg/mL of phytohemagglutinin-L (Sigma, #L2769) and 20 U/mL of interleukin-2 (BD Pharmingen, #554603) for 48 h, isolated, infected with HIV-1 *Ba-L* (MOI: $10^{-2}$) for 2 h, and cultured as previously described with the following changes [33]. Infected cells were seeded at $0.5 \times 10^6$ cells per well in 12-well plates and treated with compound/drugs pre-diluted in RPMI (2 mL final). Every 2 days, 0.5 mL of media was harvested for $p24^{CA}$ ELISA and, on day 4, media was replenished once with 1 mL of fresh RPMI (with 10% FBS, 1X Pen-Strep, and 1X Amphotericin B from Wisent Corp.) containing fresh compound and 20 U/mL of interleukin-2 (2 mL final). **Cell viability assays**. In parallel, the effect of compound/drugs on cell metabolism/proliferation were assessed by XTT assay (Sigma-Aldrich, #TOX2) while, for PBMCs, cell viability was determined by trypan blue exclusion (Invitrogen, #15250–061). For CEM-GXR cells, viable cell counts were estimated by flow cytometry (Guava HT8) and gated to cover 95% of uninfected CEM-GXR cells and 95% of viable cells in inhibition assays [44].

## Measuring expression levels of HIV-1 and host proteins

**P24 ELISAs**. P24 Gag in cell supernatants (or lysates if specified) from rtTA-HIV-Δ*Mls*-HeLa, 24ST1NLESG, or HIV-1-infected PBMCs were assayed using a HIV-1 $p24^{CA}$ Antigen Capture Assay Kit (AIDS & Cancer Virus Program, NCI-Frederick, Frederick, MD USA) or HIV-1 p24 ELISA Kit (Xpress Bio, #XB-1000) with peak Gag levels reaching ~1000, ~800, and ~6100 pg/mL, respectively, per cell type. **Western blots**. Cell lysates, containing phosphatase inhibitors (10 mM NaFl and 2 mM $Na_3VO_4$) when necessary, were prepared for immunoblotting using the following reagents. Antibodies for phospho and total MAP/MAPKs (ERK1/2, JNK1/2/3, p38α/β/γ/δ, and MAPKAPK-2), HA-tagged proteins, and $Na^+/K^+$-ATPase as well as use of Stain-Free labeling of total protein in gels were used as previously described [28]. Antibodies used for HIV-1 Gag (p24), Env (gp120), and Rev were as previously detailed [27]. Tat, tubulin, GAPDH, and isotype-specific HRP conjugated antibodies were as previously reported [33,39]. Antibodies for SR proteins include SRSF1 (Life Technologies, #324500), SRSF2 (Abcam, #ab203916), SRSF3 (Invitrogen, #33–4200), SRSF4 (using mAb104, gift from B. Chabot), SRSF6 (Novus, #NBP2-04142), SRSF7 (Abcam, #ab137247), SRSF9 (MBL, #RN081PW), and

Tra2β (Abcam, #ab31353). Proteins were resolved by reducing SDS-PAGE and transferred to PVDF membranes by a Bio-Rad Trans-Blot Turbo Transfer System or by wet electrophoretic transfer. Clarity (Bio-Rad, #170–5060) or Western Lightning ECL reagent (Perkin-Elmer, #NEL101) were used for blots bound with HRP-conjugated antibodies and signals captured by digital camera on a Bio-Rad ChemiDoc MP System or X-ray film. Unsaturated protein bands on immunoblot/gels were quantitated by ImageLab, normalized to internal loading controls (α-tubulin, GAPDH, Stain-Free labeled total protein, or β-actin), and displayed relative to DMSO (+Dox/PMA/HIV). Activation of kinases were calculated from phospho ÷ total protein levels from these data. Images acquired were rotated and exported as TIF files for assembly and equal brightness/contrast/sharpness adjustments as necessary in Microsoft PowerPoint or ImageJ. Some lanes were cropped and rearranged from the same blot/gel as indicated. In representative gel/blot sets, samples were electrophoresed from same experiments as control lanes, resolved/detected simultaneously, and transferred to same immunoblots.

## Analysis of RNAs

**Quantitation of HIV-1 RNAs**. RNAs were extracted from cells, reverse transcribed, and cDNAs analyzed by qRT-PCR to quantify HIV-1 mRNA levels (normalized to β-actin as internal loading control) as previously described for HeLa rtTA-HIV-Δ*Mls* cells [33]. qRT-PCR of RNAs from 24ST1NLESG and HeLa rtTA-HIV(Gag-GFP) cells, unless otherwise noted, used iTaq Universal SYBR Green Supermix (Bio-Rad, #172–5120) to amplify HIV-1 US/SS/MS and β-actin RNAs using a Bio-Rad CFX384 Touch Real-Time PCR Detection System on CFX Manager software with the same primer sets stated above using the following revised conditions: 95°C for 30s followed by 40 cycles of 95°C for 15s, 60°C for 15s, and 72°C for 15s [33]. Alternatively, qRT-PCRs of HIV-1 US/SS/MS and β-actin RNAs (Fig 4E) were amplified by SsoAdvanced Universal Probes Supermix (Bio-Rad, #172–5281) on a Bio-Rad CFX384 Touch using CFX Maestro software with the same primer sets using the following revised conditions: 95°C for 30s followed by 40 cycles of 95°C for 10s (or 5s) and 60°C for 20s (or 15s). **Quantification of G-protein α-subunit mRNAs**. qRT-PCR of RNAs on G-protein α subunits in HeLa rtTA-HIV(Gag-GFP) cells were amplified using iTaq and analyzed on a CFX96 Real Time PCR System or CFX96 Touch Real Time PCR System on CFX Manager using published primers (and β-actin) as detailed below via conditions as follows: 95°C for 30s followed by 40 cycles of 95°C for 20s, 53°C for 20s, and 72°C for 20s [33,81,82]. Primer sequences for Gα subunits are as follows: Gα$_{i1}$ forward (5'-AAGTACAATTGTGAAGCAGATGAAA-3'), reverse (5'-TGGTGTTACTGTAGACCACTGCTT-3'); Gα$_q$ forward (5'-GACTACTTCCCAGAATATGATGGAC-3'), reverse (5'-GGTTCAGGTCCACGAACATC-3'); Gα$_{13}$ forward (5'-TCGGGAAAAGACCTATGTGAA-3'), reverse (5'-CAACCAGCACCCTCATACCT-3'); and Gα$_s$ forward (5'-ACGTGATCAAGCAGGCTGACT-3'), reverse (5'-GGAACAGGATCACAGAGATGG-3') [81,82]. **Analysis of splice site usage in HIV-1 RNAs**. The effect of drug/compounds on AS of HIV-1 MS pre-mRNAs were analyzed by RT-PCR from cDNAs above as previously outlined [39]. **FISH**. Changes in the distribution of HIV-1 genomic/US RNAs by drug/compounds were determined in HeLa rtTA-HIV(Gag-GFP) cells using Stellaris probes comprised of a mixture of 48 Quasar 570-labeled 20-mer oligonucleotides spanning the HIV-1 Gag coding region (Biosearch Technologies) as previously detailed [27,39].

## Transcriptomics analysis

**RT-PCR**. RNAs were extracted from treated cells, reverse transcribed, and analyzed for effects on AS of cellular RNAs by medium-throughput RT-PCR (and RNA-Seq described below) as previously described [27,39,40]. The inclusion levels of 157 AS exons and splice sites located in

96 AS regions of 85 genes were assayed by an automated RT-PCR using a Biomek 2000 workstation. Fluorescent products were generated using primer sets labeled with 5-FAM. Events assayed were previously suggested to be linked to cell transformation and available for lab use. Labeled PCR products were denatured in formamide and quantitated using an ABI Prism capillary sequencer (Life Technologies). The inclusion level of each exon was calculated as the amount of transcripts carrying the alternative exon relative to the total amount of all transcripts detected in the PCR reaction. Alternative exon inclusion data (PSI) were averaged for each treatment and differences between a compound and DMSO control (ΔPSI) were calculated for each event. **RNA-Seq**. Data was obtained from poly(A)$^+$ RNA and sequenced on an Illumina HiSeq2500 as previously detailed [83]. RNA quality in samples was measured by an Agilent Bioanalyzer for a RNA integrity number (RIN) value of $\geq$ 8. For estimating AS, PSIs were determined from exon inclusion/exclusions levels in compound-treated and DMSO/control-treated cells and differences between them (ΔPSIs) calculated. From a total of 18,611 AS events counted, 9,806 events were analyzed with confidence. DE genes were determined from calculating mean fold change in the level of each gene expressed between compound and DMSO-treated cells. Mean expression levels of each gene were normalized and presented as reads per kilobase of exon model per million mapped reads (RPKM) and a threshold of RPKM $\geq$ 0.5 was applied. From a total of 19,847 genes read, 11,406 transcripts were analyzed with confidence by these parameters.

**SUnSET analysis of host total protein synthesis [56].** HeLa rtTA-HIV-Δ*Mls* cells were cultured in the presence/absence of compound and Dox induced for ~24 h to induce HIV-1 expression and then pulsed with puromycin (10 μg/mL, Sigma-Aldrich, #P8833) for 30 min to label nascent proteins prior to harvest. Cycloheximide treatment (10 μM, Sigma-Aldrich, #C4849) was used as a control. MG132 (10 μM, Sigma-Aldrich, #M7449) was added to cells 8 h prior to harvest. Cells were washed and whole cell lysates were immunoblotted by an anti-puromycin antibody (MilliporeSigma, #anti-12D10) and bands shown were quantified.

**Proteomics analysis.** LC-MS/MS analysis using quantitative TMT labeling of proteins was performed by SPARC BioCentre Molecular Analysis, The Hospital for Sick Children (SickKids), Toronto, ON, Canada [57]. **Sample preparation.** Protein samples (~1 mg) were suspended in 50 mM $NH_4HCO_3$ (pH 8.3), 10 mM DTT, heated at 60°C for 30 min, cooled to room temperature, alkylated with 10 mM iodoacetamide for 15 min in the dark, and quenched with 40 mM DTT. Proteins were digested with 1:50 (w/w) ratio of trypsin overnight at 37°C, terminated by trifluoroacetic acid (0.5%), centrifuged at 2,000 x g for 5 min, desalted by Sep-Pak C18 Vacuum Cartridges (Waters, Milford, MA), and lyophilized. For labeling, samples were reconstituted in 200 mM HEPES (pH 8.5) and incubated with TMT 10-plex isobaric quantitative-labeling reagents (Thermo Scientific) for 1 h as previously detailed, and quenched with 5% hydroxylamine [57]. Labeled peptides were pooled, desalted by Sep-Pak C18, and lyophilized. **LC/MS-MS analysis.** TMT-labeled samples were reconstituted in 5% formic acid and separated by reverse-phase chromatography on a 50 cm x 75 μm Easy-Spray column packed with 2 μm Reprosil-Pur C18-AQ resin managed by an EASY nLC 1000 chromatography system (all from Thermo Scientific). Peptides were fractionated with a 165 min linear gradient of 0–35% acetonitrile (in 0.1% formic acid, v/v) at a flow rate of 250 nL/min followed by 15 min with 35–100% acetonitrile. Peptides were subject to a nanoelectrospray ion source followed by MS/MS analysis on a Q-Exactive Hybrid Quadrupole-Orbitrap Mass Spectrometer (Thermo Scientific). Data was collected over 3 h using a chromatographic peak width of 20s, positive polarity, collision-induced dissociation (CID) of 0.0 eV, default charge state of 2, and data-dependent acquisition mode [with minimal automatic gain control (AGC) target of 5e3, intensity threshold of 5e4, charge exclusion at unassigned, 1, 8, and >8, prefer peptide matches, exclude isotopes, and dynamic exclusion of 50s]. MS1 scans for profiling spectra

were acquired over *m/z* range of 525–1800 *m/z*, 35K resolution, AGC target of 1e6, and maximum injection time (IT) of 120 ms. Selected ion monitoring in MS2 scans were applied for acquisition of top 15 most abundant MS1 ions with a resolution of 35K, CID normalized collision energy of 27%, AGC target of 5e6, maximum IT of 100 ms, isolation window of 1.6 *m/z*, scan range of 200–2000 *m/z*, and fixed first mass of 100.0 *m/z*. **Data processing by programs.** MS/MS spectra were analyzed using Sequest (version 1.4.1.14, Thermo Scientific) and X! Tandem (version CYCLONE 2010.12.01.1, The GPM) to search and match sample results of trypsin digested peptides to a human UniProt database (Aug. 8, 2016; 42,128 and 42,175 entries, resp.) concatenated with a reverse decoy database. Searches used a fragment ion-mass tolerance of 0.020 Da and a parent-ion tolerance of 10.0 ppm. TMT 10-plex addition to amino groups of lysine and N-termini of any peptide and carbamidomethylation of cysteine were specified as fixed modifications while deamidation of asparagine and glutamine, oxidation of methionine, and phosphorylation of serine, threonine, and tyrosine were specified as variable modifications. In X! Tandem, ammonia loss of the N-terminus was also specified as a variable modification. **Data analysis in programs.** Scaffold Q+ (from Scaffold, version Scaffold_4.8.7, Proteome Software Inc.) was used to validate TMT 10-plex peptide and protein identifications. Peptide and protein identifications were accepted if they could establish >93% and >99% probability, respectively, to achieve a false discovery rate (FDR, q value) of <1.0% and <1.0%, respectively (calculated using decoys with reverse protein sequences), and if proteins contained $\geq$ 2 identified peptides. Peptide probabilities from Sequest were assigned by Scaffold's local FDR algorithm while peptide probabilities from X! Tandem were assigned by the peptide prophet algorithm with Scaffold delta-mass correction. Protein probabilities were assigned by the protein prophet algorithm. Proteins that contained similar peptides and could not be differentiated based on MS/MS analysis alone were grouped to satisfy the principles of parsimony. Normalization was performed iteratively across samples and spectra on their ion intensities. Data means were averaged. Spectra data were log transformed, cut of those matching to multiple proteins or missing a reference value, and weighed by an adaptive intensity weighting algorithm. Given the current thresholds, 115,687 out of 123,895 spectra (93%) were included in the quantitation. **Analysis of exported data.** Protein abundance (relative fold change) was calculated directly from $\log_2$ normalized ion-intensity data of unfiltered modifications from Scaffold, analyzed using Perseus (version 1.6.2.2, Max-Planck Institute of Biochemistry) and Excel, and Venn diagrams drawn from http://bioinformatics.psb.ugent.be/webtools/Venn/. Proteins with substantial changes in abundance (referenced with total proteins detected) were submitted to WEB-based GEne SeT AnaLysis Toolkit (http://www.webgestalt.org/option.php) for overrepresentation enrichment analysis using the KEGG pathway database. **Data availability.** Mass spectrometry proteomics data were deposited to the ProteomeXchange Consortium (http://proteomecentral.proteomexchange.org) via the PRIDE partner repository with dataset identifiers: PXD011079 and DOI 10.6019/PXD011079.

## Monitoring intracellular signals and secondary messenger release

As described for HeLa rtTA-HIV-Δ*Mls* cells, HeLa rtTA-HIV(Gag-GFP) cells were seeded in 48-/12-well plates (6-well for RNA analyses) prior to addition of any inhibitor and Dox. Ras activation in cell lysates from HeLa rtTA-HIV(Gag-GFP) cells were measured by ELISA using a Ras GTPase activation Kit (MilliporeSigma #17–497) per manufacturer's procedures. Peak levels for controls reached ~230,000 RLUs. To determine which pathway signals inhibit HIV-1 gene expression, cells were pretreated with a specific kinase inhibitor (or knocked down) prior to addition of compound and Dox induction, and monitored for recovery of Gag-GFP expression (while kinase activation levels were determined by western blotting as described above).

Equal concentrations of DMSO were present in each experiment. Kinase inhibitors were purchased from Sigma (U0126, #U120; SB203580, #S8307; SP600125, #S5567; BAPTA-AM, #A1076), BioShop (U0126, #U0U237.5), Abcam (KB-R7943, #ab120284), Selleckchem (Selumetinib, #S1008), AOBIOUS, Inc. (BIM-46187, #AOB 6169), or MilliporeSigma (PD158780, #513036; Herbimycin A, #375670; LY294002, #440204; Calphostin C, #208725). Recombinant human EGF was from Invitrogen (#PHG0314). GFP fluorescence was initially detected in plated cells by a Typhoon Imager 9400 (Amersham Biosciences) or Typhoon FLA 9400 (GE) on ImageQuant and in cell lysates by SDS-PAGE captured digitally on a ChemiDoc MP. Before quantification, cells were washed with warm PBS and either scanned live (and/or harvested for protein/RNA analyses) or fixed in 3.7% paraformaldehyde/formaldehyde-PBS for subsequent analyses. Data from SDS-PAGE and scans for fluorescence in plated cells (described above and below) were quantitated using Image Lab and ImageJ software, respectively, and background subtracted as necessary. For detecting Gag-GFP fluorescence in plates, cells were washed once and read in warm PBS on a Typhoon Imager 9400 as detailed: 488 nM excitation, 526 SP (or 520 BP) emission filter, ~600–665 PMT, +3 mm focal plane, 100 μ, and normal sensitivity, or on a Typhoon FLA 9400: 473 nM excitation, LPB emission filter, and ~765–865 PMT. Changes in $[Ca^{2+}]_i$ and ROS in treated cells were monitored by Fura Red AM and CellROX Deep Red Reagent (#F-3020 and C10422, resp., Life Technologies) using a Typhoon FLA 9400 following manufacturer's instructions. For detecting $[Ca^{2+}]_i$, cells in 48-well plates were loaded with 10 μM of Fura Red AM for 1 h at 37C, washed once and read in 200 uL of warm PBS as detailed: 488 nM excitation, 670 BP emission filter, ~600 PMT, +3 mm focal plane, 100 μ, and normal sensitivity. For ROS, cells in 48/96-well plates were loaded with 5 μM of CellROX Deep Red for 0.5 h at 37C, washed 3 times with PBS, and read in 200 uL of warm PBS as detailed: 633 nM excitation, 670 BP emission filter, ~525 PMT, +3 mm focal plane, 100 μ, and normal sensitivity. In parallel, the density of cells treated with various inhibitor combinations were assayed by methylene blue stain (2% in 50% ethanol, BioBasic, #MB0342) by incubating fixed cells for $\geq$ 0.5 h, wash 2–3 times with ddH2O, and read at $OD_{664}$ on a TECAN Infinite 200 PRO or Biotek Cytation5.

## Exogenous expression of small G proteins

HeLa rtTA-HIV(Gag-GFP) cells were seeded a day prior, washed twice with PBS, serum starved in IMDM/DMEM for 3.5 h, and transfected in Opti-MEM (Invitrogen, #31985070) with equal amounts of plasmid DNA [pCGN-HA-N-Ras (WT, 12D, or 17N) or CMV-Myc-3xTerm] with/without tetracycline transactivator (CMV-tTA-pA or CMV-Myc-3xTerm) and placental alkaline phosphatase (CMV-Myc-PLAP) using a 1:3 ratio (w/v) of 1 μg/uL of polyethylene imine (PEI) 25K (Polysciences Inc., #23966–2) diluted in Opti-MEM [60]. After ~4 h, cells were replaced with fresh Opti-MEM containing 1X antibiotics and, after 48 h, washed twice with PBS prior to harvest for analysis of protein/RNAs.

## Statistical analyses

Data were analyzed using Microsoft Excel (or GraphPad Prism) and expressed as means ± standard error of the mean (s.e.m.). Differences between two groups of data, i.e. drug/compound treatment vs. control (DMSO +Dox/PMA/HIV), were compared by two-tailed Welch's/Student's *t*-test for each target indicated. For intracellular signaling graphs, cells pretreated with no kinase inhibitor (Ctr) and a compound were compared to those with no kinase inhibitor (Ctr) and DMSO (+) per target indicated (illustrated by a gray-dashed line on one of the targets) whereas cells pretreated with a kinase inhibitor and a compound (e.g. 5342191) within one treatment set were compared to those with no kinase inhibitor (Ctr) and

a compound within the same set per target indicated (illustrated by a black-dashed line on one of the targets). Alternatively, MG132 replaces "kinase inhibitor" described above. Statistical significance in results are indicated on graphs for each p value as follows: $p < 0.05$, *; $p < 0.01$, **; and $p < 0.001$, ***. For proteomic/transcriptomics, Perseus provided multiple- and two-sample testing by ANOVA and *t*-test, respectively, and Benjamini-Hochberg procedure (and/or by modifying spreadsheet from www.biostathandbook.com/multiplecomparisons.html) for false discovery rates (FDRs).

## Supporting information

**S1 Table. RNA-Seq dataset presenting genes with significant changes in alternative splicing in cells treated with 5342191.**
(XLSX)

**S2 Table. RT-PCR dataset showing the genes with significant differences in alternative splicing in cells treated with 5342191.**
(XLSX)

**S3 Table. RNA-Seq dataset displaying significant changes in gene expression in cells treated with 5342191.**
(XLSX)

**S4 Table. Relative abundance of 5,326 proteins quantified by TMT LC-MS/MS analysis from cells treated with 5342191, 9147791, or DMSO +/- Dox.**
(XLSX)

**S5 Table. Proteins identified in cells with substantial alterations in abundance from treatment with 5342191, 9147791, or DMSO without (-) Dox.**
(XLSX)

**S6 Table. Categories of pathways enriched among proteins with substantial changes in abundance from cells treated by different compounds.**
(XLSX)

**S1 Fig. Pattern of HIV-1 mRNA products generated from splicing.** Illustrated is the organization of the HIV-1 proviral genome (top) indicating the position of multiple 5' splice donor sites (SD1-4) and 3' splice acceptor sites (SA1-7) used in the splicing of viral pre-mRNA. Below is a diagram of the alternatively spliced RNAs generated by processing HIV-1 genomic RNA [unspliced (US), 9 kb]. Indicated are the common (open boxes) and alternative exons (closed boxes) used in generating the singly spliced (SS, 4 kb) and multiply spliced (MS, 1.8 kb) viral RNAs (bottom) and the nomenclature used to describe the exon composition of each mRNA generated from these two classes of HIV-1 RNAs. Note that two isoforms of Tat are translated from these exons: p14 Tat from SS mRNAs and p16 Tat from MS mRNAs. SS mRNAs generate a truncated form of Tat (p14) due to the presence of a termination codon immediately 3' of SD4, producing the shorter isoform. The mRNA for *env* is also bicistronic, encoding *env/vpu* because of an additional *vpu* open reading frame (ORF) upstream of the *env* ORF.
(TIF)

**S2 Fig. Gel/blots used for representative figures.** Lanes from continuous and unexcised gel/blots were cropped and rearranged for Fig 1D (**A**) and 1E (**B**), Fig 2I and 2J (**C-D**), S5 Fig (**E**), S6 Fig (**F**), S7A Fig (**G**), S11A Fig (**H**), S11B (**I**), S11C Fig (**J**), S11D Fig (**K-L**), and S13C Fig (**M**).
(TIF)

**S3 Fig. RT-PCR and RNA-Seq data demonstrate that 5342191 alters a small subset of alternatively spliced host RNAs.** (**A**) A total of 70 alternative splicing events were analyzed by RT-PCR of cDNAs from HeLa rtTA-HIV-Δ*Mls* cells treated with 2 μM of 5342191 or DMSO (control) per Fig 1 and quantitated by capillary electrophoretic sequencing to determine the levels of alternative exon inclusion (PSI; S2 Table, n = 3, mean). To display differences, mean PSIs from cells treated with 5342191 (y-axis) were plotted versus cells treated with DMSO (x-axis). ΔPSIs of events which were significantly different between 5342191 and DMSO treated cells (p <0.05) were indicated with colored circles as follows: <10% (black), 10–20% (red), and ≥ 20% (yellow, with gene identity shown). (**B**) Alternative splicing in cells quantified by RT-PCR in (**A**, x-axis) correlate with those from RNA-Seq (y-axis, S1 Table and Fig 2E and 2F). Of ΔPSIs quantified, a total of 17 alternative splicing events were compared and their strength of correlation (Pearson) was determined (r = 0.83).
(TIF)

**S4 Fig. Changes in cell viability from exposure of HeLa cervical carcinoma cells to 5342191.** HeLa rtTA-HIV-Δ*Mls* cells were treated with 2 μM of 5342191 (191, purple diamonds) or DMSO (control, black circles) per Fig 1 and cell viability monitored by XTT assay over a course of 4 days as indicated (n ≥ 3, mean, s.e.m.).
(TIF)

**S5 Fig. Effect of 5342191 on the expression of SR proteins.** HeLa rtTA-HIV(Gag-GFP) cells were treated with 2.5 μM of 5342191 or DMSO control and Dox (+) induced per Fig 2I–2K. Cell lysates (~30 μg) were analyzed for changes in SR protein expression by immunoblotting with antibodies specific for SRSF 2, 7, or 9, or Tra2β in parallel with SR proteins blotted in Fig 2I–2K. Blots are representative of n ≥ 3 experiments and quantified in graph shown in Fig 2K. Stain-Free-labeled total proteins served as internal loading control and for normalization of these data. Lanes were cropped and assembled from the same gel (S2E Fig). Note: the lower amount of protein observed in lane 3 does not represent a change in SR protein levels after normalization of this data with total protein detected and graphed in Fig 2K.
(TIF)

**S6 Fig. Effect of 5342191 on splice site usage of HIV-1 MS pre-mRNAs.** (**A-C**) HeLa rtTA-HIV-Δ*Mls* cells were treated with 2 μM of 5342191 and cDNAs (from RNAs extracted and reverse-transcribed as described in Fig 2B) were analyzed by RT-PCR of the MS RNA class (n ≥ 3–4, mean, s.e.m.). (**A**) Illustration indicating the position of primers used for PCR (arrow heads, see S1 Fig for depiction of the products generated). (**B**) Representative gel and (**C**) graph of the MS mRNA species (x-axis) quantified from RT-PCR amplification and displayed as a percentage (%) of the total HIV-1 MS mRNAs (y-axis). Lanes in (**B**) were cropped and assembled from the same gel (S2F Fig).
(TIF)

**S7 Fig. 5342191 addition to cells does not affect nascent synthesis of host proteins.** HeLa rtTA-HIV-ΔMls cells were treated with/without 2 μM of 5342191 and Dox per Fig 1 and nascent protein synthesis in cells monitored by SUnSET. (**A-B**) Representative immunoblots and (**C**) graph quantifying results from nascent peptides detected by anti-puromycin antibody in lysates of treated cells (~40 μg) as indicated (n ≥ 3, mean, s.e.m.). Unlabeled cells and the translation inhibitor, cycloheximide (CHX), were used as negative controls. Results are relative and statistically compared to DMSO (+). Blot in (**A**) was cropped and assembled from S2G Fig.
(TIF)

**S8 Fig. Volcano plots illustrating significant and differentially expressed proteins in cells treated with different compounds.** $Log_2$ relative fold change ($\Delta$) in protein abundance of HeLa rtTA-HIV-$\Delta Mls$ cells treated with (**A**) 2 μM 5342191 (n = 3), (**B**) 30 μM 9147791 (n = 3), or (**C**) DMSO without Dox (-, n = 2) from DMSO (+) reference were plotted against $-log_{10}$ of their corresponding p values to identify significant and substantial changes among 5,326 proteins identified from TMT LC-MS/MS (S4 Table). A red-dashed horizontal line displays the cut-off for significance at p <0.05 (1.30) while two red vertical lines display the threshold for 1.5-fold $\Delta$ (+/- 0.585). Proteins with <1.5-fold change (+/- 0–0.58) are shown with gray circles, 1.5-2-fold $\Delta$ (+/- 0.59–0.99) as yellow boxes, and >2-fold $\Delta$ (+/- $1.0^+$) as green triangles. Proteins deemed both statistically significant (p <0.05) and substantially changed are presented as diamonds by the same color scheme for 1.5-2-fold change (yellow) and >2-fold change (green) and provided in S5 Table.
(TIF)

**S9 Fig. Scatter plots and percentage scores displaying differences in protein abundance of cells treated with different compounds.** A total of 5,326 proteins identified by TMT LC-MS/MS (from S4 Table) were analyzed from HeLa rtTA-HIV-$\Delta Mls$ cells treated with 30 μM of 9147791 (n = 3), 2 μM of 5342191 (n = 3), or DMSO without Dox (-, n = 2) as described and compared with data in Fig 3C and 3D. DMSO with Dox (+) served as the control/reference for these results. (**A-B**) Averaged normalized ion intensity ($log_2$) from (**A**) 9147791 or (**B**) DMSO (-) treated cells (y-axes) were plotted against DMSO (+, x-axes) to illustrate differences in the abundance of proteins. Results are depicted by gray circles for <1.5-fold change ($\Delta$, +/- 0–0.585), orange circles for 1.5-2-fold $\Delta$ (+/- 0.59–0.99), and green circles for >2-fold $\Delta$ (+/- $1.0^+$). A Pearson correlation (r) and a blue-dashed theoretical line representing a drug with no effect on protein abundance are shown with these data. (**C-D**) Total count in the number and percentage (%) of proteins affected by (**C**) 9147791 or (**D**) DMSO (-) tallied based on their relative fold $\Delta$ from DMSO (+). Protein abundances that were substantially affected by these treatments (**A-D**) were listed in S5 Table and any overlap in proteins affected in common with these different treatments and 5342191 were depicted by a Venn diagram in Fig 3E. (**E-F**) Venn diagram comparing the changes observed between transcriptomics (RNA-Seq) and proteomics (TMT LC-MS/MS) in 5342191-treated cells. DE genes with $\geq$ 2-fold change were compared with proteins with abundances with $\geq$ 1.5-fold (**E**) and 1.3-fold change (**F**).
(TIF)

**S10 Fig. Temporal analysis of 5342191 treatment on HIV-1 gene expression.** HeLa rtTA-HIV(Gag-GFP) cells were treated for 0.5, 1, 4, and 24 h with 5342191 (1.9 μM) or DMSO (control). After indicated time (except 24 h), cells were washed and fresh media containing equal concentrations of DMSO and Dox added. Cells were harvested after a total of 24 h and HIV-1 Gag-GFP expression quantified by detecting GFP fluorescence from cells as described in S12 Fig. Assay was performed in triplicate and results displayed as mean, s.e.m, and relative to DMSO (+).
(TIF)

**S11 Fig. MEK1/2-ERK1/2 activation is required for 5342191 suppression of HIV-1 gene expression.** HeLa rtTA-HIV(Gag-GFP) cells were pretreated with/without a kinase inhibitor prior to treatment with 2 μM 5342191 or DMSO and Dox per Fig 4 to determine the signaling pathway(s) involved by monitoring kinase activation and/or rescue of Gag-GFP expression in cells. Cells were pretreated with/without a kinase inhibitor (and confirmed to block each kinase) as follows: MEKi #1 (12 μM U0126), MEKi #2 (5 μM Selumetinib), JNKi (1.25 μM SP600125), or p38i (15 μM SB203580) overnight for ~15 h (**B-C** and **G**, Fig 4A, and/or

alongside previous study); [Ca2+]i (5 µM BAPTA-AM) or NCXi (5 µM KB-R7943) for 3 h (Fig 4B); and Gαi (~10 µM BIM-46187,), EGFRi (120 nM PD158780,), Srci (350 nM Herbimycin A), or PI3Ki (10 µM LY294002) for 3 h (via activity of Ras in Fig 5A, EGF in S12E Fig, and/ or ERK1/2 in **G**). Impact of each inhibitor combination on cell density was monitored in S12B, S12D and S12F Fig. Tubulin or Stain-Free-labeled total proteins served as internal loading controls. (**A-C**) Representative immunoblots quantitating activation levels of ERK1/2 without (**A**) or with MEKi #1 (**B**) or #2 (**C**), JNK1/2/3 (**A**), MK-2 (**A**), and p38 (**A**) graphed in Fig 4A from 35, 20, and 20 µg of cell lysates (resp., representative of n $\geq$ 3–5). (**C-F**) Representative gels quantifying Gag-GFP expression by GFP fluorescence on reducing SDS-PAGE (and initially from plated cell scans in S12A and S12C Fig) graphed in Fig 4C and 4D from 20, 35, 20, and 20 µg of cell lysates (resp., representative of n $\geq$ 3–4). (**G**) Western blot quantifying ERK1/ 2 activation levels with/without MEKi #2 or Gαi in cell lysates (20 µg). Lanes in (**A-C** and **D**, resp.) were cropped/assembled from same simultaneously run blot/gels (S2H–S2J and S2K– S2L Fig) and (**G**) was run/detected simultaneously. Samples in (**C**) were prepared in p24$^{CA}$ ELISA sample diluent containing 1% BSA (66.5 kDa).
(TIF)

**S12 Fig. Suppression of HIV-1 gene expression by 5342191 involves MEK1/2-ERK1/2 signaling and activation of G proteins.** HeLa rtTA-HIV(Gag-GFP) cells were pretreated with/ without a kinase inhibitor prior to addition of 1.9 µM of 5342191 or DMSO (control) and Dox as described and analyzed in Fig 4. To identify the signaling pathway(s) used by 5342191 to inhibit HIV-1 gene expression, treated cells were assayed for GFP fluorescence in plates for rescue of Gag-GFP expression (**A**, **C**, and **E**) and cell density determined by methylene blue stain (**B**, **D**, and **F**). Cells were pretreated with (**A-B**) an inhibitor of MEK1/2 (MEKi #1, 12 µM U0126), p38α/β/β2 (p38i, 15 µM SB203580), or JNK1/2/3 (JNKi, 1.25 µM SP600125) overnight for ~15 h (n $\geq$ 6–10 and 6–10, resp., mean, s.e.m.), (**C-D**) intracellular Ca$^{2+}$ chelator ([Ca2+]i, 5 µM BAPTA-AM) or NCX Ca$^{2+}$ influx inhibitor (NCXi, 5 µM KB-R7943) for ~3 h (n $\geq$ 3–10 and 3, resp., mean, s.e.m.), or (**E-F**) pan Gα subunit (Gαi, ~10 µM BIM-46187), ErbB/HER (EGFRi, 120 nM PD158780), Src (Srci, 350 nM Herbimycin A), or PI3K inhibitor (PI3Ki, 10 µM LY294002; n $\geq$ 4–8 and 5–6, resp., mean, s.e.m.). EGF (50 ng/mL) was added in (**E**) as a control for activating HIV-1 gene expression via EGFR/PI3K signaling. Results in graphs (**A**, **C**, and **E**) were confirmed by SDS-PAGE analysis of Gag-GFP expression in Fig 4C and 4D and the inhibitory activity of each kinase inhibitor was confirmed in Figs 4A and 4B and 5A, (**E**), and/or alongside our previous study. Results are shown relative to DMSO (+) with no kinase inhibitor. Statistical comparisons were performed as illustrated by gray/black-dashed lines (and asterisks) for one of the targets and described in Materials and methods.
(TIF)

**S13 Fig. Treatment of cells with 5342191 has no significant effect on the levels of ROS and Na$^+$/K$^+$-ATPase.** HeLa rtTA-HIV(Gag-GFP) cells were treated with 5342191 (1.9 µM) or DMSO (control) and induced by Dox (+) or left uninduced (-) as described in Fig 1. (**A**) Graph of ROS levels monitored by CellROX Deep Red labeling of treated cells (n $\geq$ 3, mean, s. e.m.). (**B**) Representative blot and (**C**) graph quantifying the level of Na$^+$/K$^+$-ATPase in cell lysates (30 µg) by immunoblots (n $\geq$ 3, mean, s.e.m.). Tubulin served as internal loading control and for normalization of this data. Lanes in (**B**) were cropped and assembled from the same blot (S2M Fig).
(TIF)

**S14 Fig. Treatment of cells with leptomycin B (LB) blocks expression of US and SS HIV-1 mRNAs.** HeLa rtTA-HIV-Δ*Mls* cells were treated with 20 ng/mL of LB and HIV-1 RNAs

harvested for quantitation by qRT-PCR of HIV-1 US (white), SS (gray), and MS (black) mRNAs as described in Fig 2A and 2B (n $\geq$ 3, mean, s.e.m.). Results of each HIV-1 RNA class are shown relative and statistically compared to each DMSO (+) control as described in Materials and methods.
(TIF)

**S15 Fig. Illustration of the intracellular signaling pathways involved in 5342191 inhibition of HIV-1 gene expression.** This model, expanded from Fig 5E, proposes that 5342191 inhibits HIV-1 gene expression in a similar manner as CSs through activation (highlighted yellow) of MEK1/2-ERK1/2 signaling. In contrast to CSs, which bind to the Na$^+$/K$^+$-ATPase, 5342191 initiates this signaling cascade (red-colored targets) through activation of G$\alpha$ subunits initiated by GPCRs at the cell membrane. Although JNK and MK-2 are activated by 5342191, these have limited/no effect on HIV-1 gene expression (blue-colored targets). Additionally, p38 MAPK, ROS, and [Ca$^{2+}$]$_i$ flux, which were not activated by this compound, were highlighted in gray while inhibition of PI3K-AKT, EGFR, and Src kinase signals that have limited/no effect on rescuing HIV-1 gene expression in the presence of 5342191 were labeled blue. Although 5342191 activates JNK and MK-2 like CSs, it does not induce MK-2 in a p38-dependent manner or [Ca$^{2+}$]$_i$ flux. In line with these signaling events, several downstream genes (*italicized*) were also significantly activated upon 5342191 treatment of cells. Signaling molecules that were not tested for activation or recovery of HIV-1 gene expression in the presence of 5342191 were left unchanged (unhighlighted as gray-colored targets) while ones with inconclusive data were labeled black. Exogenous expression of signaling targets ($^\dagger$) or an inhibitor with cytoxic effects ($^*$) were also noted in the diagram.
(TIF)

## Acknowledgments

We thank the following people and their institutions for assistance. SickKids: J. Tong of M. Moran's lab for assistance on LC-MS/MS data analysis, C.A. Lingwood and Nades Palaniyar for work space (and ChemiDoc MP use), and K. Johnson-Henry and S. Robinson of P. M. Sherman's lab and K. Hogarth of J. T. Mayne's lab for CFX96 usage; University of Toronto: M. Ohh and Y. Kano for N-Ras plasmids, S. Gray-Owen lab and blood donors for PBMCs, and S. Hyde and S. Campbell of L. Cowen's lab for CFX384 use; and Canadian Blood Services: B. Binnington for proofreading a cover letter.

## Author Contributions

**Conceptualization:** Raymond W. Wong, Alan Cochrane.

**Data curation:** Raymond W. Wong.

**Formal analysis:** Raymond W. Wong, Ahalya Balachandran, Peter K. Cheung, Qun Pan, Alan Cochrane.

**Funding acquisition:** Raymond W. Wong, Ahalya Balachandran, Alan Cochrane.

**Investigation:** Raymond W. Wong, Ahalya Balachandran, Peter K. Cheung, Ran Cheng, Peter Stoilov.

**Methodology:** Raymond W. Wong, Ahalya Balachandran.

**Project administration:** Raymond W. Wong, Alan Cochrane.

**Resources:** Raymond W. Wong, Peter Stoilov, P. Richard Harrigan, Benjamin J. Blencowe, Donald R. Branch, Alan Cochrane.

**Supervision:** Raymond W. Wong, P. Richard Harrigan, Benjamin J. Blencowe, Donald R. Branch, Alan Cochrane.

**Validation:** Raymond W. Wong, Ahalya Balachandran, Peter K. Cheung, Alan Cochrane.

**Visualization:** Raymond W. Wong.

**Writing – original draft:** Raymond W. Wong.

**Writing – review & editing:** Raymond W. Wong, Alan Cochrane.

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
