## [Decision Letter · Decision Letter 0]

12 Nov 2019

Dear Dr. Wong:

Thank you very much for submitting your manuscript "An activator of G protein-coupled receptor and MEK1/2-ERK1/2 signaling inhibits HIV-1 replication by altering viral RNA processing" (PPATHOGENS-D-19-01870) for review by PLOS Pathogens. Your manuscript was fully evaluated at the editorial level and by independent peer reviewers. The reviewers appreciated the attention to an important topic but identified some aspects of the manuscript that should be improved.

We therefore ask you to modify the manuscript according to the review recommendations before we can consider your manuscript for acceptance. Your revisions should address the specific points made by each reviewer.

(1) A letter containing a detailed list of your responses to the review comments and a description of the changes you have made in the manuscript. Please note while forming your response, if your article is accepted, you may have the opportunity to make the peer review history publicly available. The record will include editor decision letters (with reviews) and your responses to reviewer comments. If eligible, we will contact you to opt in or out.

(2) Two versions of the manuscript: one with either highlights or tracked changes denoting where the text has been changed; the other a clean version (uploaded as the manuscript file).

We hope to receive your revised manuscript within 60 days or less. If you anticipate any delay in its return, we ask that you let us know the expected resubmission date by replying to this email.

[LINK]

Sincerely,

Edward M Campbell, PhD

Guest Editor

PLOS Pathogens

Thomas Hope

Section Editor

PLOS Pathogens

Kasturi Haldar

Editor-in-Chief

PLOS Pathogens

orcid.org/0000-0001-5065-158X

Grant McFadden

Editor-in-Chief

PLOS Pathogens

orcid.org/0000-0002-2556-3526

Dr. Cochrane,

All three reviewers have read your revised manuscript and find it much improved. They do have some minor comments that I feel still require attention. Of these, I would not insist that you address reviewer 3's question about the effect being upstream or downstream of Rev. Although I agree with them that this is an important question, given the scope of the study in its current form I would leave this important mechanistic question for future studies by your group. I would however ask you to address reviewer 1's concerns on the western blot in figure 1D, as it is an important early piece of data providing support for the premise of the study. If you could respond to this comment with an improved western blot and address the other comments of the reviewers, which are largely editorial questions, I would be happy to receive a revised copy of your manuscript.

Reviewer's Responses to Questions

**Part I - Summary**

Reviewer #1: The authors have significantly improved the delivery of their study by rewriting of the manuscript text and improving the data presentation. They have answered the critiques appropriately. Minor points remain to be addressed.

The manuscript revision presents a cogent analysis of an inhibitor of HIV mRNA processing. The drug appears to invoke a sensitive response by HIV that diminishes the balanced synthesis of structure and regulatory proteins to attenuate virus replication. The study uses state of the art approaches to measure generalized effect on host mRNAs and specific effect on HIV mRNAs. The text cogently synthesizes a vast dataset into a tenable model.

Reviewer #2: The revised manuscript by Wong et al has addressed my major points. The authors show that 5342191 inhibits HIV-1 replication at concentrations that do not cause detectable cytotoxicity. The mechanisms by which this drug works are complicated, but it appears to act by activating MEK/ERK signaling pathways. This drug may be a useful tool to study how signaling pathways regulate HIV-1 gene expression.

Reviewer #3: This is a resubmission of a manuscript from Wong et al. describing a small molecule inhibitor of HIV gene expression called 5342191 (aka 191). The study is impressive in its breadth- showing the drug to suppress HIV gene expression in several cell types without major effects on cell viability; and featuring large-scale comparative host cell transcriptome and proteome analyses. The major conclusion is that the drug alters HIV alternative splicing through a yet to be resolved mechanism, potentially through modulating the levels or activities of SR family proteins. The authors had previously attributed the splicing effects to 191 triggering MEK/ERK signaling through Galpha and N-Ras. However, based on new data they walk back these claims to the point where they now argue that 191 triggers one of potentially multiple complex signaling pathways that link GCPRs and HIV-1 splicing program. Overall the authors have done a good job of responding to reviewers' comments and re-working a lot of interpretations. There are remaining mechanistic gaps and a persisting, unnecessary desire to argue that the drugs don't affect cell physiological; but overall the paper is improved.

**Part II – Major Issues: Key Experiments Required for Acceptance**

Reviewer #1: There is need to make equivalent loading control in figure 1D. Lane 3 underloaded compared to lane 2, making it ineffective to compare the HIV protein signals between the -/+ drug treatment.

Reviewer #2: None

Reviewer #3: Figs 2 and 3. It’s still strange that Tat and Rev expression is abolished by 191 if the host proteome is largely intact (and seems unlikely its degradation…). It still seems to be a key question as to what is happening to Rev and Tat despite high levels of MS transcripts, and if loss of Rev really underpins the overall defect. This is handled better in the discussion than before, but can the authors resolve if 191 is acting upstream or downstream of Rev?

**Part III – Minor Issues: Editorial and Data Presentation Modifications**

Reviewer #1: P9 typo .........with the exception of SRSF1 (~2-fold decrease)… with the exception of SRSF1 (~50% NOT 2-fold decrease), .........and SRSF3 (~30% decrease),

P30 top, first word correction to: use

Instead of ….used replaces “kinase inhibitor”

Figure 1D/ Gag WB: Gapdh exposure + DMSO (lane 2) is >>>than Gapdh +DMSO plus 5312191 (lane 3). The lanes require equivalent loading.

Need to address the following:

From PMID: 30118183…..Glycogen Synthase Kinase-3 (GSK-3) is a highly conserved negative regulator of receptor tyrosine kinase, cytokine, and Wnt signaling pathways. GSK-3 regulates alternative splicing in response to T-cell receptor activation, and recent phosphoproteomic studies have revealed that multiple splicing factors and regulators of RNA biosynthesis are phosphorylated in a GSK-3 dependent manner. The inhibition of GSK-3 alters the splicing of hundreds of mRNAs, indicating a broad role for GSK-3 in the regulation of RNA processing. GSK-3 regulated phosphoproteins include SF3B1, SRSF2, PSF.

1. Since GSK-3 is a critical regulator of mRNA processing, how do the kinase inhibitors used in this study affect GSK3 substrates?

2. Since PSF is known to regulate balanced splicing of HIV-1 (PMID: 29846681), what is the impact of this 5312191on PSF activity and how much is the contribution to the reduced replication?

I suggest the title is further improved to focus on the primary aim: Targeting HIV mRNA processing efficiency is a valid strategy for antiviral drug development.

Note: An entire page of the Introduction text is summarizing the study, which seems excessive and unwarranted and redundant.

Reviewer #2: In Figure 4E, is the color code correct? The SS and MS colors appear reversed on the figure relative to the legend.

Reviewer #3: 1. General- still some uses of diminutives (“little/no”, “minor”, “minimal”, etc.) in referring to changes to host processes in response to 191 (P3, P5, P8, P10, P17); again, considering big effects on MEK/ERK, SR proteins, cell proliferation, etc., and discernible changes in big data, seems inappropriate and unnecessary to use these terms.

2. In general, many experimental details useful to the reader (nature of assay, drug concentration, timing, etc.) are still lacking throughout the results section.

3. Abstract- S3- mean “total cell protein synthesis”; S4- still seems uncertain that US lack of transport to cytoplasm is due to loss of Rev. S5- no evidence that Tat levels are due to degradation effects.

4. Figs. 1B and 1C. Why are the X-axes not equivalent for these experiments?

5. Fig. 1G. The authors suggest an IC50 for HIV gene expression of 750 nM based on Figs. 1B and 1C but saw no effect at this concentration over six days in their replication experiment. Please resolve.

6. Fig. 1H-I. Would be best if consistent and show concentrations of drug on X-axes. Also, if the effects are due to splicing then the assay is not measuring LTR activity. Please correct.

7. Fig. 2B, C, etc. The authors argue in their response that RNA levels should be normalized to the Dox or PMA conditions but seems more useful to the reader to normalize to the negative or baseline.

8. Fig. 5D. Problem here with the labels? Should be US SS MS?

9. Fig S2. While it is good to see the original blots included, they are not very interpretable with only cherry-picked labels.

10. Fig S5 seems to show a major 191-driven loss to SRSF2, SRSF7, SRSF9, and Tra2beta expression that is not reflected in Fig 2K. Please explain/resolve.

PLOS authors have the option to publish the peer review history of their article (what does this mean?). If published, this will include your full peer review and any attached files.

Reviewer #1: No

Reviewer #2: No

Reviewer #3: No

---

## [Editor Report · Decision Letter 1]

6 Jan 2020

Dear Dr. Wong,

We are pleased to inform that your manuscript, "An activator of G protein-coupled receptor and MEK1/2-ERK1/2 signaling inhibits HIV-1 replication by altering viral RNA processing", has been editorially accepted for publication at PLOS Pathogens. 

Before your manuscript can be formally accepted and sent to production, you will need to complete our formatting changes, which you will receive by email within a week. Please note that your manuscript will not be scheduled for publication until you have made the required changes.

IMPORTANT NOTES

(1) Please note, once your paper is accepted, an uncorrected proof of your manuscript will be published online ahead of the final version, unless you’ve already opted out via the online submission form. If, for any reason, you do not want an earlier version of your manuscript published online or are unsure if you have already indicated as such, please let the journal staff know immediately at plospathogens@plos.org.

(2) Copyediting and Proofreading: The corresponding author will receive a typeset proof for review, to ensure errors have not been introduced during production. Please review the PDF proof of your manuscript carefully, as this is the last chance to correct any errors. Please note that major changes, or those which affect the scientific understanding of the work, will likely cause delays to the publication date of your manuscript. 

(3) Appropriate Figure Files: Please remove all name and figure # text from your figure files. Please also take this time to check that your figures are of high resolution, which will improve the readbility of your figures and help expedite your manuscript's publication. Please note that figures must have been originally created at 300dpi or higher. Do not manually increase the resolution of your files. For instructions on how to properly obtain high quality images, please review our Figure Guidelines, with examples at: http://journals.plos.org/plospathogens/s/figures.

(4) Striking Image: Please upload a striking still image to accompany your article if one is available (you can include a new image or an existing one from within your manuscript). Should your paper be accepted, this image will be considered for our monthly issue image and may also appear on our website to feature your article. Please upload this as a separate file, selecting "striking image" as the file type upon upload. Please also include a separate "Other" file with a caption, including credits and any potential copyright information. Please do not include the caption in the main article file. If your image is from someone other than yourself, please ensure that the artist has read and agreed to the terms and conditions of the Creative Commons Attribution License at http://journals.plos.org/plospathogens/s/content-license. Please note that PLOS cannot publish copyrighted images.

(5) Press Release or Related Media: If your institution or institutions have a press office, please notify them about your upcoming paper at this point, to enable them to help maximize its impact. If they will be preparing press materials for this manuscript, please inform our press team in advance at plospathogens@plos.org as soon as possible. We ask that you contact us within one week to plan ahead of our fast Production schedule. If you need to know your paper's publication date for related media purposes, you must coordinate with our press team, and your manuscript will remain under a strict press embargo until the publication date and time. This means an early version of your manuscript will not be published ahead of your final version. 

(6)  PLOS requires an ORCID iD for all corresponding authors on papers submitted after December 6th, 2016. Please ensure that you have an ORCID iD and that it is validated in Editorial Manager.  To do this, go to ‘Update my Information’ (in the upper left-hand corner of the main menu), and click on the Fetch/Validate link next to the ORCID field.  This will take you to the ORCID site and allow you to create a new iD or authenticate a pre-existing iD in Editorial Manager

(7) Update your Profile Information: Now that your manuscript has been provisionally accepted, please log into Editorial Manager and update your profile, if needed. Go to https://www.editorialmanager.com/ppathogens, log in, and click on the "Update My Information" link at the top of the page. Please update your user information to ensure an efficient production and billing process. 

(8) LaTeX users only: Our staff will ask you to upload a TEX file in addition to the PDF before the paper can be sent to typesetting, so please carefully review our Latex Guidelines http://journals.plos.org/plospathogens/s/latex in the meantime.

(9) If you have associated protocols in protocols.io, please ensure that you make them public before publication to guarantee immediate access to the methodological details.

Best regards,

Edward M Campbell, PhD

Guest Editor

PLOS Pathogens

Thomas Hope

Section Editor

PLOS Pathogens

Kasturi Haldar

Editor-in-Chief

PLOS Pathogens

orcid.org/0000-0001-5065-158X

Michael Malim

Editor-in-Chief

PLOS Pathogens

orcid.org/0000-0002-7699-2064
---

## [Editor Report · Acceptance letter]

7 Feb 2020

Dear Dr. Wong,

We are delighted to inform you that your manuscript, "An activator of G protein-coupled receptor and MEK1/2-ERK1/2 signaling inhibits HIV-1 replication by altering viral RNA processing," has been formally accepted for publication in PLOS Pathogens.

Best regards,

Kasturi Haldar

Editor-in-Chief

PLOS Pathogens

orcid.org/0000-0001-5065-158X

Michael Malim

Editor-in-Chief

PLOS Pathogens

orcid.org/0000-0002-7699-2064